# A two-step merging strategy for incorporating multi-source precipitation products and gauge observations using machine learning classification and regression over China

Huajin Lei [1], Hongyu Zhao [2], Tianqi Ao [1]

[1] State Key Laboratory of Hydraulics and Mountain River Engineering, College of Water Resource and Hydropower, Sichuan University, Chengdu 610065, China;

[2] State Key Laboratory of Earth Surface Processes and Resource Ecology, Beijing Normal University, Beijing 100875, China;

*Correspondence to*: Tianqi Ao(aotianqi@scu.edu.cn)

**Abstract.** Although many multi-source precipitation products (MSPs) with high spatio-temporal resolution have been extensively used in water cycle research, they are still subject to various biases, including false alarm and missed bias. Precipitation merging technology is an effective means to alleviate this uncertainty. However, how to efficiently improve precipitation detection efficiency and precipitation intensity simultaneously is a problem worth exploring. This study presents a two-step merging strategy based on machine learning (ML) algorithms, including gradient boosting decision tree (GBDT), extreme gradient boosting (XGBoost), and random forest (RF). It incorporates six state-of-the-art MSPs (GSMaP, IMERG, PERSIANN-CDR, CMORPH, CHIRPS, and ERA5-Land) and rain gauges to improve the accuracy of precipitation from precipitation identification and estimation during 2000-2017 over China. Multiple environment variables and spatial autocorrelation are combined in the merging process. The strategy first employs classification models to identify wet and dry days and then combines regression models to predict precipitation amounts based on classified wet days. The merged results are compared with traditional methods, including multiple linear regression (MLR), ML regression models, and gauge-based Kriging interpolation. A total of 1680 (70%) rain gauges are randomly chosen for model training and 692 (30%) for performance evaluation. The results show that: (1) The multi-sources merged precipitation products (MSMPs) outperformed all original MSPs in terms of statistical and categorical metrics, which substantially alleviates the bias in temporal and spatial. The modified Kling-Gupta efficiency (KGE), critical success index (CSI), and Heidke Skill Score (HSS) of original MSPs have been improved by 15-85%, 17-155%, and 21-166%, respectively. (2) The spatial autocorrelation plays a significant role in precipitation merging, which considerably improves the model accuracy. (3) The performance of MSMPs obtained by the proposed method is superior to MLR, Kriging interpolation, and ML regression models. XGBoost algorithm is more recommended for large-scale data merging owing to its high computational efficiency. (4) The two-step merging strategy performs better when higher density gauges are used to model training. But it has strong robustness and can also obtain better performance than original MSPs even when the gauges number is reduced to 10% (237). This study provides an accurate and reliable method to improve precipitation accuracy under complex climatic and topographic conditions. It could be applied to other areas well if rain gauges are available.

# 1    Introduction

As one of the critical parameters of the natural water cycle, precipitation helps us realistically understand the interaction between hydrological and climate systems. Meanwhile, precipitation monitoring is essential for extreme hydroclimatic disaster forecasting and water resources management (Yilmaz et al., 2005; Tao et al., 2016; Xu et al., 2018). Accurate precipitation estimates are of practical importance for social economy and security, agriculture, meteorology, ecology, and other fields (Awange et al., 2019). Traditional rain gauge measurements can provide reliable precipitation data. It only reflects the precipitation characteristics within a limited radius around the instruments (Collischonn et al., 2008; Jia et al., 2011). The distribution of gauges is scarce and irregular, particularly in Tibetan Plateau where this study is covered and where precipitation has significant spatiotemporal variability (Ma et al., 2021). Mapping precipitation spatial patterns based on gauges observations may cause large uncertainties. In contrast, satellite-based precipitation estimates and atmospheric reanalysis are attractive alternative tools for describing spatial continuous distribution due to their high spatio-temporal resolution.

Up to the present, a series of advanced remote sensing techniques and numerical weather models have been employed to retrieve various multi-source precipitation products (MSPs) (Huffman et al., 2007; Joyce et al., 2004). For instance, the Tropical Rainfall Measuring Mission (TRMM) algorithm combines detection information from multiple sensors (including the microwave imager, infrared radiometer, and radar) to provide valuable precipitation information for tropical and subtropical regions (Huffman et al., 2007). The Climate Hazards Group InfraRed Precipitation with Station data (CHIRPS) (Funk et al., 2015) incorporates infrared cold cloud duration observations and satellite information to prepare a long time and high spatial resolution (0.05°) dataset. The Precipitation Estimation from Remotely Sensed Information using Artificial Neural Networks (PERSIANN) applies a state-of-the-art algorithm to generate global precipitation based on the geostationary longwave infrared imagery (Hsu et al., 1997). As an extension of TRMM, the Integrated Multi-satellitE Retrievals for GPM (IMERG) algorithm enhances the estimation efficiency of solid and light precipitation, which has finer temporal resolution and wider spatial coverage than TRMM (Huffman et al., 2019). In addition to satellite-based precipitation products, the National Centers for Environment Prediction and National Center for Atmospheric Research (NCEP/NCAR) and the European Centre for Medium Range Weather Forecasts (ECMWF) have yielded many reanalysis products, such as ERA-Interim, NCEP/NCAR, and ERA5. The latest ERA5-Land provides a variety of land climate variables over serval decades with an enhanced spatial resolution compared to ERA5 (Hersbach et al., 2020). Nevertheless, previous studies have already demonstrated that MSPs usually suffer from various degrees of uncertainty caused by retrieval algorithms, complex terrain, limitation sensors resampling frequency, and assimilation techniques (Nerini et al., 2015; Arshad et al., 2021; Xu et al., 2022). This uncertainty tends to be more severe at shorter time scales (such as sub-daily and daily) and varies among different precipitation products (Lei et al., 2021). Therefore, how alleviating the errors of MSPs is a crucial priority step to improve their application efficiency (Jiang et al., 2012; Sharifi et al., 2016; Lu et al., 2020).

An important means to improve the accuracy of MSPs is to combine multi-source products and gauge-based precipitation information. In this way, the deficiencies caused by a single or independent data source could be compensated (Xie and Arkin, 1997; Nie et al., 2015). The widely used statistical methods include optimal interpolation (OI) (Xie and Xiong, 2011; Shen et al., 2014; Wu et al., 2018), quantile mapping (QM) (Piani et al., 2010a; Katiraie-Boroujerdy et al., 2020; Tong et al., 2021), geographically weighted regression (GWR) (Chao et al., 2018; Chen et al., 2020), inverse-root-mean-square-error weighting (Shen et al., 2014; Yang et al., 2017), one-outlier removed (OOR) (Shen et al., 2014), Bayesian model averaging (Ma et al., 2017; Yumnam et al., 2022), geographical difference analysis (GDA) (Duan and Bastiaanssen, 2013; Arshad et al., 2021), Kriging-based method (Manz et al., 2016), and multi-method coupled (Wu et al., 2018; Lu et al., 2020). Although the aforementioned approaches have obtained better performance in some regions, they are strongly based on solid mathematical assumptions and suffer various limitations (Wu et al., 2020). For example, the QM method removes biases in the statistical periods but cannot capture precipitation wet/dry day lengths and interannual variability (Ajaaj et al., 2015). The OOR method simply calculated the weight by the linear average of all values (Ma et al., 2017). Most importantly, these statistical methods are difficult to describe the relationship between the precipitation process and complex environmental variables (Shen et al., 2014; Wu et al., 2018).

The rapid development of machine learning (ML) technology can overcome some limitations caused by the above methods. Compared to traditional approaches, ML can deal with complex nonlinear relationships without constructing explicit statistical models. Moreover, the strength of ML comes from its ability to solve different types of problems, from classification to regression and prediction, as well as its efficiency in learning and generalizing massive amounts of data (He et al., 2016). Those features make various ML methods extensively adopted in precipitation calibration and merging. Such as random forest (RF) (Baez-Villanueva et al., 2020; Chen et al., 2021), quantile regression forest (QRF) (Bhuiyan et al., 2018, 2019), support vector machine (SVR) (Kumar et al., 2019), convolutional neural network (CNN) (Le et al., 2020), deep neural network (DNN) (Tao et al., 2016), artificial neural networks (ANN) (Wehbe et al., 2020; Hong et al., 2021), long-short-term memory network (LSTM) (Tang et al., 2021; Yang et al., 2022), as well as multi-algorithms coupling (Wu et al.,2020; Tan et al., 2021; Zhang et al., 2021). However, most above studies mainly considered limited environmental information and spatial correlation related to precipitation while neglecting the spatial autocorrelation between gauge observations in merging processes. For example, the Euclidean distance in Baez-Villanueva et al. (2020), geographical coordinates, and inverse distance weighted (IDW) in Zhang et al. (2020). In addition, the uncertainty of MSPs is partly caused by unsatisfactory precipitation identification, which not only influences the statistical length and start/end time of wet/dry days, but further leads to the overestimation/underestimation of precipitation intensity. Correctly judging whether precipitation events occur is the key to enhancing precipitation performance fundamentally. Several studies have employed ML methods to discriminate precipitation/non-precipitation, such as Zhang et al. (2021) used SVM, RF, ANN, and extreme learning machine, Tao et al.

(2016) and Xiao et al. (2022) applied ANN, and Pham et al. (2019) used RF and SVM. However, those studies incorporated gauge observations with several MSPs or a single source. Each product has its pros and cons, and sufficient products should be considered to extract valuable information (Zhang et al., 2021, Lei et al., 2022). In addition, to the best of our knowledge, the gradient boosting decision tree (GBDT) and extreme gradient boosting (XGBoost) algorithms have not been well explored in precipitation discriminating and merging.

To address above mentioned concerns, this study proposes a two-step merging strategy to incorporate six popular MSPs (one latest reanalysis and five satellite products) and relatively high-density rain gauges over China from 2000-2017, focusing on enhancing the precipitation discrimination ability and absorbing MSPs' strengths. This strategy is based on XGBoost, GBDT, and RF classification and regression models, and multiple environmental information especially spatial autocorrelation are taken into consideration. The objectives of this study mainly include three-folds: (1) exploring the effectiveness of the proposed

strategy in all aspects according to various metrices; (2) comparing the performance of the proposed strategy with traditional methods; (3) assessing the influence of MSPs' spatial resolution and gauge density on model performance. This strategy is expected to improve the accuracy of existing MSP and explore the potential of more ML algorithms in precipitation.

## 2 Study area and Materials

### 2.1 Study area

China, between 73°-135°E and 15°-53°N, is selected as the study area, which is located in eastern Asia and west of the Pacific Ocean with a land area of 9.6 million km$^2$ (Fig. 1). The elevation of China gradually increases from southeast to northwest, resulting in a complex topography including mountains, plateaus, hills, basins, and plains. China has a diverse climate, including temperate monsoon climate, subtropical monsoon climate, tropical monsoon climate, temperate continental climate, and plateau mountain climate. Tibetan plateau is dominated by the plateau mountain climate with a low temperature,

strong radiation, abundant sunshine, and little precipitation. However, the southern region has a subtropical monsoon climate characterized by warm winter, hot summer, and abundant rainfall. Annual precipitation over China has high spatial variability, varying between 50 mm and 2000 mm from west to east. Meanwhile, the distribution of precipitation amounts and events throughout the year is also extremely uneven. Much more precipitation (70% - 80%) occurs during the warm season (May to October) than during the cold season (November to April), which is the primary factor for this study to conduct model training

according to different seasons. In addition, China is mainly divided into nine river basins, from east to south, including Continental basin (CB), Songliao river basin (SLRB), Yellow river basin (YERB), Haihe river basin (HARB), Southwest basin (SWB), Yangtze river basin (YARB), Huaihe river basin (HURB), Southeast basin (SEB), and Pearl river basin (PRB) (Fig. 1). The runoff of most basins mainly comes from precipitation, while CB is mainly from snow and glacier meltwater.

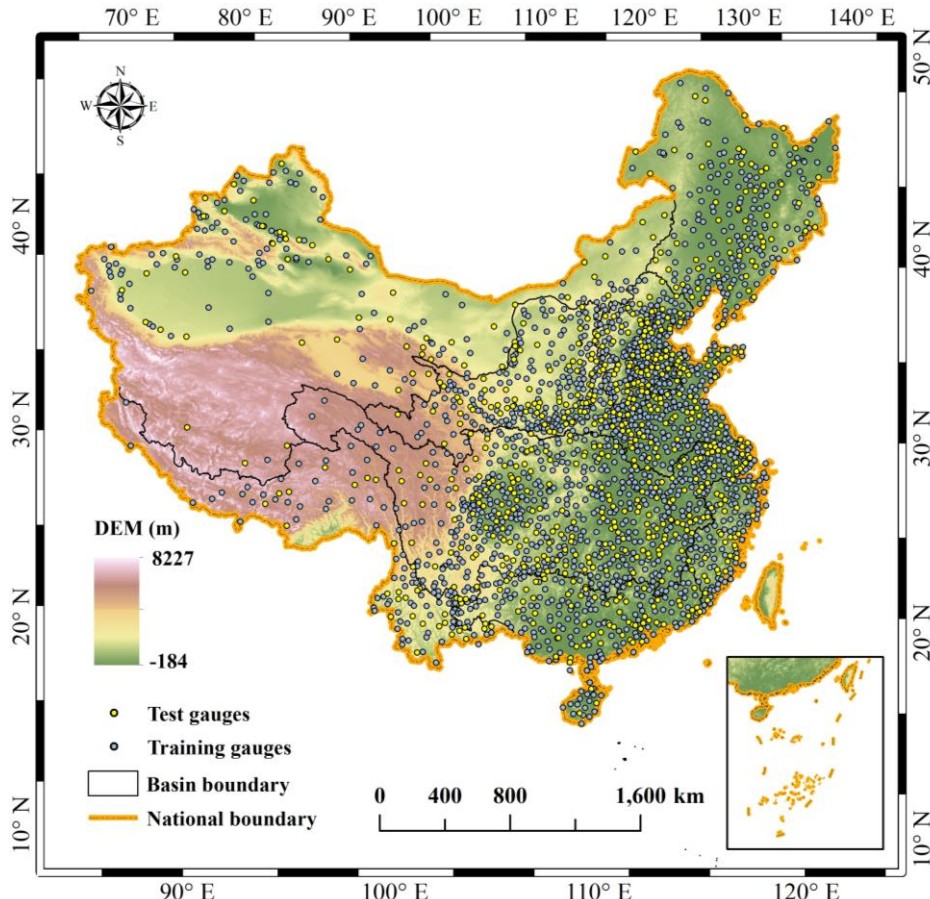

**Figure 1: The topography of China and distribution of rain gauges.**

## 2.2 Materials

### 2.2.1 Rain gauge observations

A relatively dense network of 2372 rain gauges over mainland China from 2000 to 2017 is collected in this study, provided by China Meteorological Administration (CMA). The daily precipitation data have been conducted strictly quality control by CMA. These quality control processes include removing extreme values, internal consistency check, and spatial consistency check (Shen et al., 2010). Therefore, gauges can be used after simple processing, such as converting units. It should be noted that there is a temporal mismatch (12h) between daily gauge-based precipitation (Beijing Time from 20:00 to 20:00, UTC + 8:00) and MSPs (UTC, from 00:00 to 24:00). Considering that not all products have a sub-daily scale temporal resolution, we recalculate daily observations using sub-daily precipitation (i.e., 8:00 to 20:00 and 20:00 to 8:00) to keep consistent with MSPs. Gauges are mainly distributed in eastern but sparsely located in western China, especially in the hinterland of Qinghai-Tibet Plateau (TP) (as shown in Fig. 1). The gauge density used in this study is higher than in some previous studies (Wu et al., 2020; Yin et al., 2021; Zhang et al., 2021). The average control area for a single gauge is approximately 4000 km$^2$ (9.6×10$^6$ km$^2$/2372). Nevertheless, it is far from meeting the requirement of the World Climate Organization that the control area should be about 600 km$^2$ for plain and even smaller for mountains regions (WMO, 1965).

### 2.2.2 MSPs

Six continuously updated products are selected to integrate, including a reanalysis product and five satellite precipitation products retrieved from multiple sensors. Limited by the availability of MSPs, the period of this study is from June 2000 to December 2017 (hereafter: 2000-2017). Specific information about MSPs is summarized in Table 1.

IMERG is the level 3 product of Global Precipitation Measurement (GPM) algorithm. The IMERG algorithm incorporates the multi-source information from the GPM microwave imager, Visible and Infrared Radiometer (VIRS), and space-borne Ku/Ka-band dual-frequency radar. IMERG provides three types of products, including Early, Late, and Final Run products, which are retrieved around 4h, 12h, and 4month, respectively, after satellite monitoring. The IMERG Final run product outperforms the Early and Late because it combines the Global Precipitation Climatology Centre (GPCC) gauge observations. The latest version 6 Final run product is therefore chosen in this study. Moreover, the Global Satellite Mapping of Precipitation (GSMaP) GSMaP_Gauge applied in this study incorporates Climate Prediction Center (CPC) gauge data analysis (Kubota et al., 2007), which is more accurate than other GSMaP products such as GSMaP near-real-time (NRT).

PERSIANN-Climate Data Record (PERCDR) has a long record from 1983 to the present. The PERSIANN algorithm is mainly based on Gridded Satellite (GridSat-BI) IR data and National Centers for Environmental Prediction (NCEP) Stage IV radar data (Ashouri et al., 2015), which does not fuse microwave information. The reliability of PERCDR is improved by using GPCC for calibration. CHIRPS v.2 product is also used in this study. It has higher spatial resolution than other MSPs, integrating satellite imagery, global climatology, and gauge observations. In addition, Climate Prediction Center Morphing Technique (CMORPH) version 1 dataset (Joyce et al., 2004) covers three categories' products: CMORPH RAW, CMORPH bias-corrected (CRT), and CMORPH gauge blended datasets (BLD). CMORPH CRT is selected in this study due to its superior quality.

ERA5-Land (herein ERA5L) is an enhanced land atmospheric reanalysis dataset of the fifth generation ERA5 produced by ECMWF. It provides various land surface variables for more than 70 years with continuous updates. ERA5L describes the evolution of the water and energy cycles on the land in a consistent manner (Hersbach et al., 2020). ERA5L adopts cycle 41r2 of ECMWF's Integrated Forecast System (IFS). Compared with ERA5 and older ERA-Interim, ERA5L employed a better 4-dimensional variational (4D-var) assimilation technique, with an enhanced horizontal resolution (9km) and higher spatial resolution (0.1°). As one of the art-of-the-art reanalysis data, ERA5L has been widely used in many fields (Xin et al., 2021; Xu et al., 2022).

The information sources employed in MSPs show significant differences, especially whether microwave signals are incorporated or not (Table 1). Moreover, various algorithms are adopted to retrieve precipitation in different MSPs. For instance, the Kalman filtering technique is employed for GSMaP, the Goddard Profiling Algorithm 2014 is used for IMERG, and the morphing technique is applied for CMORPH (Table 1). Each algorithm and signal source has its cons and pros. It is necessary

to combine them to maximize their advantages. Although several products already combine gauge observation data (e.g., GPCC and CPC) to reduce bias, only a few gauges within China are used. Given the relatively high gauge density used in this study, this has little impact on the independence of gauges and the reliability of results (Shen et al., 2013). The number and location of gauges used in GPCC over China is shown in Appendix A.

**Table 1. The information about seven MSPs used in this study**

| MSPs | Temporal-spatial resolution | Spatial coverage | Input sources | Retrieval algorithm |
|------|------|------|------|------|
| GSMaP | 1h, 0.1° | 60°S-60°N | PMW, IR and Gauge | Kalman filtering technique |
| IMERG | 0.5h, 0.1° | 60°S-60°N | PMW, IR and Gauge | Goddard Profiling Algorithm |
| PERCDR | 3h, 0.25° | 60°S-60°N | IR and Gauge | adaptive ANN |
| CHIRPS | daily, 0.05° | 50°S-50°N | IR, Gauge, and reanalysis | Kalman filter model |
| CMORPH | 3h, 0.25° | 60°S-60°N | PMW, IR and Gauge | Morphing technique |
| ERA5L | 1h, 0.1° | Global | Reanalysis and Gauge | IFS Cy41r2 4D-Var |

### 2.2.3 Environment variables

The environment variables used in this study include DEM, longitude, latitude, wind speed, relative humidity, soil moisture, cloud cover, and air temperature.

DEM is downloaded from the Shuttle Radar Topographic Mission (SRTM) with a resolution of 90 m. Wind speed, relative
humidity, soil moisture, and air temperature are obtained from the NASA Global Land Data Assimilation System Noah Land Surface Model (GLDAS_NOAH), with 3 h and 0.25° resolutions (Rodell., 2004). Cloud cover is collected from ERA5 because it is not included in GLDAS_NOAH, with the resolution of hourly and 0.25°. Although Normalized Differential Vegetation Index (NDVI) is often used as a critical auxiliary variable to predict precipitation, it is susceptible to soil type and human activities. NDVI is more suitable for monthly or annual applications due to its temporal resolution (Ghorbanpour et al.,
2021; Shen et al., 2021; Tan et al., 2021). Inversely, the response of air temperature and soil moisture to daily precipitation is better than NDVI, especially in the desert and bare land (Bhuiyan et al., 2018). In addition, the interactions between cloud properties and precipitation are equally important (Sharifi et al., 2019).

## 3    Methodology

### 3.1 Data preprocessing

In this study, the period of model training and precipitation interpolation are from 2000 to 2017 at the daily scale. To maintain data's temporal and spatial consistency, all MSPs and environment variables at a sub-daily scale are aggregated to daily data. The spatial resolution of DEM (90m) and CHIRPS (0.05°) are upscaled to 0.1°, the PERCDR, CMORPH, cloud cover, and GLDAS_NOAH are downscaled to 0.1° using the bilinear interpolation method. In this study, the gauges are divided into two groups, 70% of rain gauges (1680) are spatially and randomly selected as training and calibrating samples, and the

remaining 30% (692) as validation samples. Due to the irregular distribution of rain gauges over China, random sampling is

carried out for each river basin to ensure the spatial representativeness of the validation gauges.

    Inspired by previous researches (Baez-Villanueva et al., 2020; Zhang et al., 2020), we consider a covariate describing

spatial autocorrelation between rain gauges in this study. The semivariogram based on Ordinary Kriging is adopted to calculate

spatial autocorrelation factor, i.e., Kriging_based prediction (KP). Compared with other predict models, such as Inverse

distance interpolation (IDW), the Kriging_based semivariogram considers not only the spatial relationship between predicted

and neighboring known points but considers the statistical autocorrelation between known points. The Ordinary Kriging

assumes the model:

$$z^*(x_0) = \sum_{i=1}^{n} \lambda_i z(x_i),$$ (1)

Where $z^*(x_0)$ is the predicted value of the unknown $x_0$ point. $z(x_i)$ and $\lambda_i$ are the known value of neighboring rain gauge $x_i$ and

its weight. Unbiasedness and minimum estimation variance are the conditions for choosing weights. The weight depends on

the distance between the known points, the predicted position, and the overall spatial arrangement based on the known points.

Spatial autocorrelation must be quantified before spatial arrangement can be applied in weights. The calculation processes of

KP are as follows:

(1) Calculate the distance and semivariogram between known points;

$\gamma(h) = \frac{1}{2}[z(x_i) - z(x_j)],$ (2)

    Where $\gamma(h)$ is the semivariogram of $x_i$ and $x_j$, $h$ is the distance, $z$ is the value of known of points.

(2) A theoretical model is used to fit semivariogram and distances. The nugget, sill, and range can be obtained according to

the fitted semivariogram. The commonly used semivariogram models are spherical, exponential, Gaussian, and linear

models. Compared with the prediction performance of KP by different models, the spherical model with better

performance was selected in this study. For more information about comparison results, refer to the Appendix B. The

spherical model is as follows:

$$\gamma(h) = \begin{cases} 0 & h = 0 \\ C_0 + C\left(\frac{3}{2} \cdot \frac{b}{a} - \frac{1}{2} \cdot \frac{b^3}{a^3}\right) & 0 < h \le a \\ C_0 + C & h > a \end{cases},$$ (3)

    Where $\gamma(h)$ is semivariogram, $h$ is the distance, $C_0$, $C$, and $a$ is the nugget, sill, and range, respectively.

(3) Calculate the semivariogram between the unknown point and known points, and form a matrix to solve the weights:

$$\begin{bmatrix} \gamma(h_{11}) & \cdots & \gamma(h_{1n}) & 1 \\ \vdots & \ddots & \vdots & \vdots \\ \gamma(h_{n1}) & \cdots & \gamma(h_{nn}) & 1 \\ 1 & \cdots & 1 & 0 \end{bmatrix} \cdot \begin{bmatrix} \lambda_1 \\ \vdots \\ \lambda_n \\ \mu \end{bmatrix} = \begin{bmatrix} \gamma(h_{10}) \\ \vdots \\ \lambda_n \\ \gamma(h_{n0}) \end{bmatrix},$$ (4)

    Where $\mu$ is Lagrange parameter.

(4) Predict the value of the unknown point using eq. (1) according to the weights obtained from eq. (4).

## 3.2 A two-step merging strategy

The specific process of the two-step merging strategy is illustrated in Fig.2. The random forest (RF), gradient boosting decision tree (GBDT), and extreme gradient boosting (XGBoost) are chosen to incorporate six MSPs (GSMaP, IMERG, PERCDR, CMORPH, CHIRPS, and ERA5L) and rain gauges. Although the RF method has been extensively employed in most previous studies, few studies compared it with GBDT and XGBoost models in precipitation merging. The environment variables, including soil moisture, cloud cover, relative humidity, air temperature, DEM, longitude, latitude, and spatial autocorrelation (KP) are selected as auxiliary variables (i.e., covariate) of the merging step1 and step2. The values of multiple covariables and MSPs extracted according to gauge locations are taken as independent variables, while gauge observations are taken as the dependent variable. Meanwhile, according to the annual distribution characteristics of precipitation, we group all input datasets into two seasons: warm season (May and October) and cold season (November to April), and models are trained independently in each season.

The two-step merging strategy explored in this study can be generally described in two stages (Fig. 2) as follows:

(1) Precipitation classification. The biases of precipitation products mainly come from overestimating/underestimating the amounts of hit events and failing to correctly distinguish precipitation occurrence, including false alarm and missed events (Lei et al., 2022). Therefore, the first step aims to classify precipitation to reduce the missed and false alarmed bias. The gauge observations are distinguished to wet/dry days according to the 0.1mm/d threshold value (Lei et al., 2021; Yu et al., 2020; Jiang et al., 2021) and used as the benchmark for classification. The wet day is set as 1 and the dry day is set as 0. The feature values of MSPs and covariables corresponding to each grid are applied to construct XGBoost, GBDT, and RF classification models. The model determines whether a day in the grid is a wet day or a dry day according to the classification probability. Hence, the classification result contains only wet and dry days (0,1) of each grid and does not involve precipitation intensity. In addition, the model is constructed in warm and cold seasons using divided independent datasets, which leads to six classification models (i.e., two seasons with three models).

(2) Precipitation regression. Precipitation regression focuses on improving the precipitation intensity of hit events. The MSPs and covariables values corresponding to the wet day of gauge observations are extracted, which are used to construct and train XGBoost, GBDT, and RF regression models. Similarly, six regression models are trained. The trained regression models are then applied to predict precipitation amounts of wet days (value equals 1) classified in step (1), while dry days remain 0. The final multi-source merged precipitation products (MSMPs) are obtained by predicted in each grid and day prediction. MSMPs in the whole period are derived from the combination of cold and warm seasons, which are named PXGB2, PGBDT2, and PRF2 respectively according to different models.

To highlight the superiority of the two-step merging strategy, we compare it with single ML regression, multiple linear regression (MLR), and gauge-based Kriging interpolation methods. Meanwhile, the best-performing algorithm is selected by

intercambiaring the three ML models in the two-step merging strategy. The detailed merging algorithms are introduced in 3.2.1-

3.2.4.

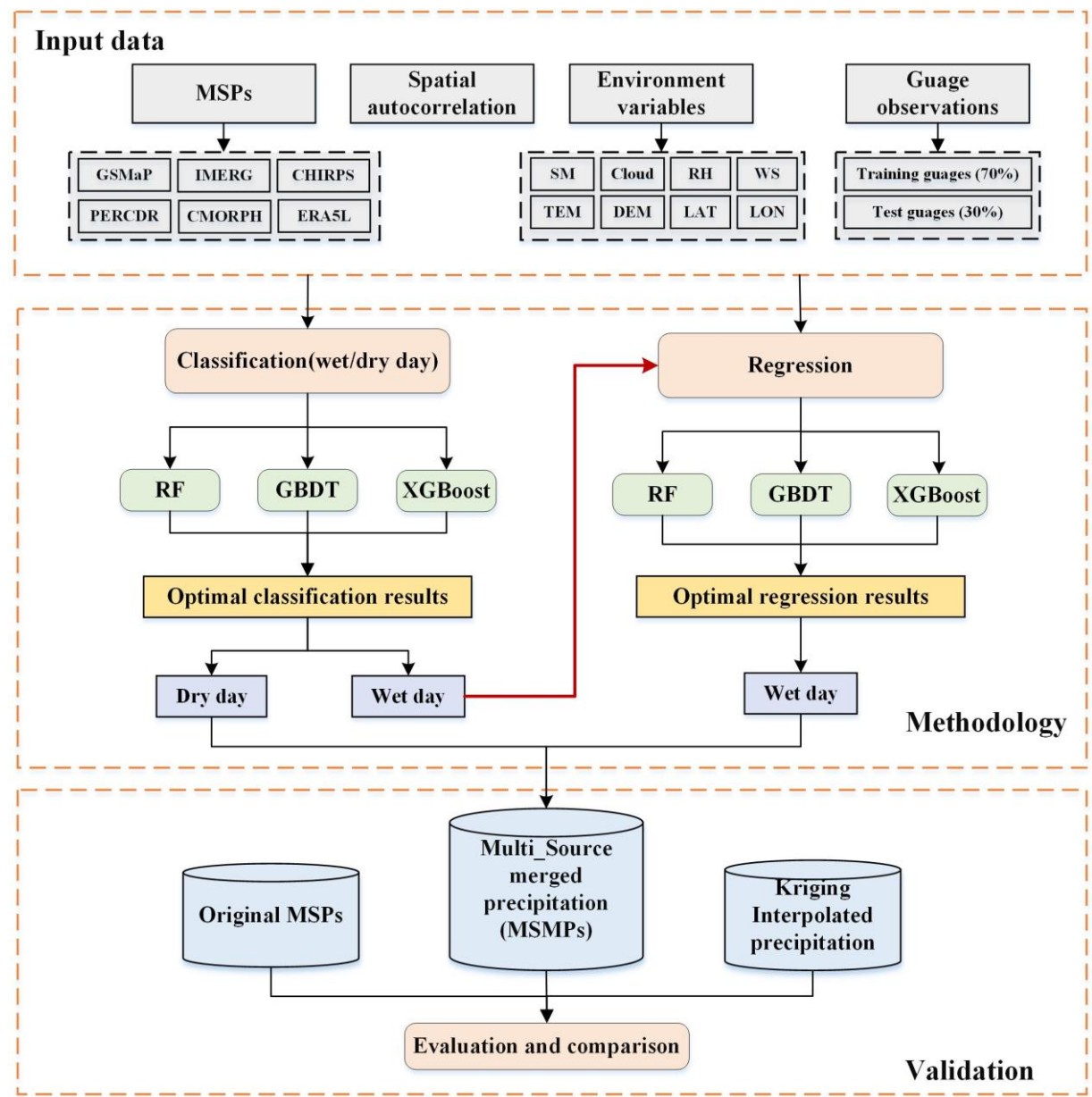

**Figure 2. The flowchart of merging strategy of this study (LAT is latitude, LON is longitude, RH is relative humidity, SM is soil moisture, TEM is temperature, and WS is wind speed).**

### 3.2.1 RF

The RF model was proposed by Breiman (2001) and is widely applied to deal with regression, classification, and other tasks (Rodriguez-Galiano et al., 2012; Nguyen et al., 2021). The general structure of RF is shown in Fig. 3. RF is an ensemble learning algorithm composed of multiple decision trees and generally outperforms a single tree. For regression problems, the model returns predictions by averaging all individual decision trees. For classification problems, each tree in the forest is judged and classified separately, and the output of RF is the class of a majority vote on classification trees (Ho, 1998).

The Bootstrap Aggregation (i.e., Bagging) technique is applied by the RF training algorithm for tree learners, which is

designed to improve the accuracy and stability of ML algorithms in classification and regression processes. The Bagging algorithm utilizes the out-of-bag (OOB) error to measure the prediction error of RF. It creates two independent datasets. One dataset, the Bootstrap sample (approximately two-thirds of all samples), is selected as "in-the-bag" data through sampling and replacement, while the remaining out-of-bag dataset (one-third) that is not selected during the sampling process is used to

calculate the model's OOB error (Breiman, 2001). The advantages of RF can be mainly summarized in four points: (1) processing high-dimensional data (a mass of features) without dimensionality reduction and feature selection; (2) measuring the importance of features and how they interact with each other; (3) avoiding overfitting and easy to implement; (4) balancing errors for asymmetric datasets, which is critical in the cold season when wet and dry days are unevenly distributed. In addition, several important parameters in RF are the number of decision trees (n_estimators), the maximum depth of each decision tree

(max_depth), and the minimum number of samples required to split an internal node (min_samples_split). A trial-and-error procedure is used to optimize model parameters due to the large sample size used in this study (approximately 14 million pieces of data) and the limitation of computing resources. The optimal parameters of model training during the warm season and cold season is displayed in Appendix C.

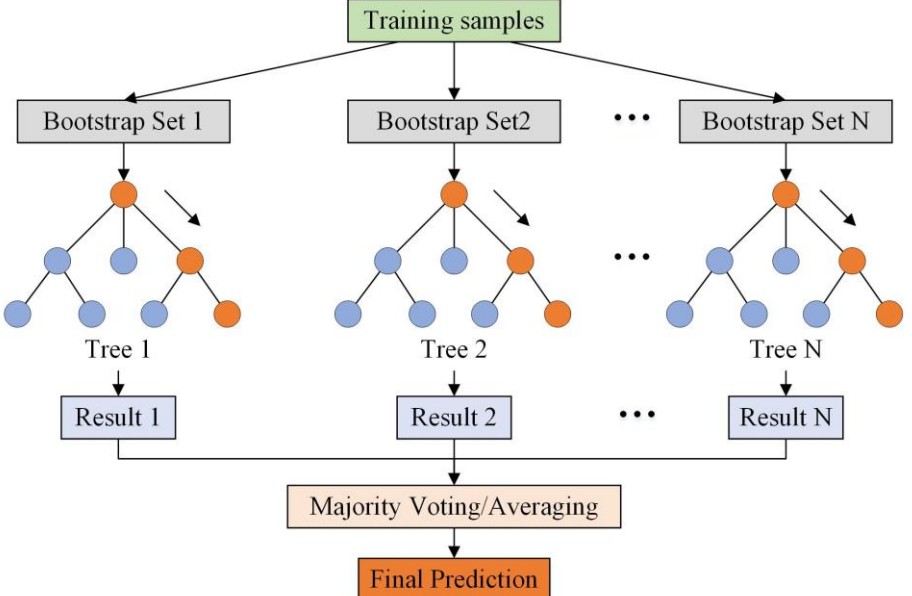

**Figure 3. The overview structure of RF.**

**3.2.2 GBDT**

The GBDT is an iterative decision tree model created by Breiman (1997) and subsequently developed by Friedman (2002), which is also called the multiple additive regression tree (MART) (shown in Fig. 4). The additive algorithm is utilized for classification or regression to continuously reduce residuals generated in the training process. GBDT uses the forward

distribution algorithm and selects the classification and regression tree (CART) learner as a weak base learner. GBDT generates numerous weak learners through multiple iterations, and each learner is trained based on the residual of the previous learner. It finally integrates the multiple weak learners into a single strong learner by weighting the summation of each tree.

The main difference between RF and GBDT is that RF can be trained in parallel to reduce variances, while GBDT reduces the biases by fitting the residual of former trees. Due to the strong connection between weak learners, GBDT is difficult to be paralleled. Generally speaking, GBDT has superior generalization ability and robustness, which is less affected by training samples size and can deal with various data flexibly, including outliers and irrelevant features. Moreover, the prediction accuracy of GBDT is high in the case of relatively little parameter adjustment time. The main parameters of GBDT include the number of boosting stages to perform (n_estimators), the learning rate shrinks the contribution of each tree by learning_rate (learning_rate,) and the maximum depth of trees (max_depth). The n_estimators and learning rate are highly correlated with the performance of the model. The optimal parameters are shown in Appendix C.

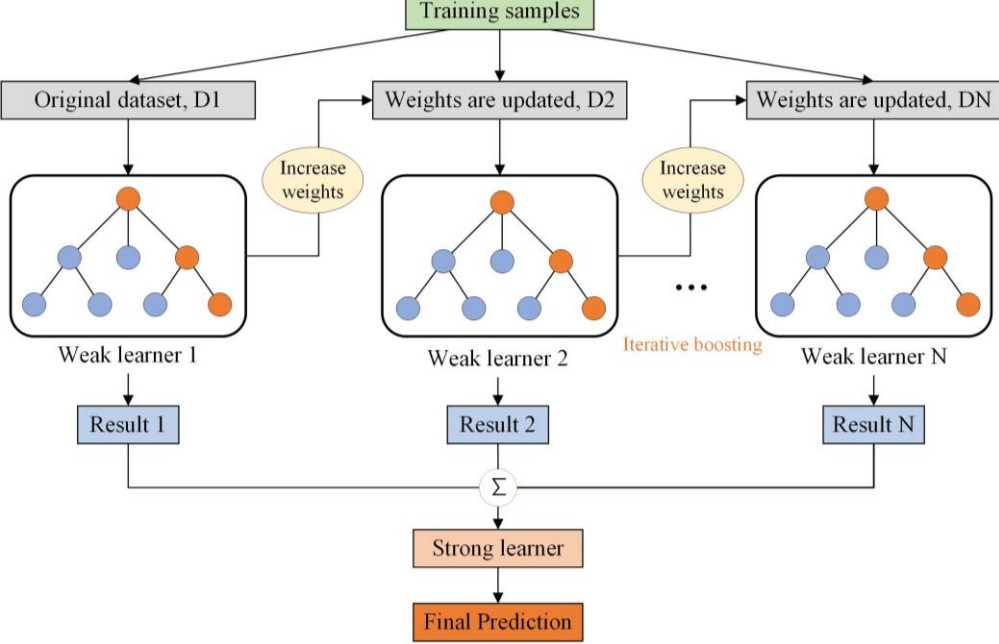

**Figure4. The overview structure of GBDT.**

### 3.2.3 XGBoost

The XGBoost model was proposed by Chen and Guestrin (2016) based on the structure of GBDT. XGBoost also combines multiple weak learners into a strong one, and the base learner in XGBoost can be either CART or linear classifier. XGBoost possesses the strength of GBDT and has several additional improvements: First, GBDT only uses the first-order derivative information in optimization, while XGBoost performs second-order Taylor expansion on the cost function to obtain the first-order and second-order derivatives, thus acquiring more accurate loss functions. Second, XGBoost introduces a regularization term into the cost function to effectively control the complexity of the model. From the perspective of bias-variance tradeoff, it reduces the variance of the model, making the learned model more straightforward and preventing over-fitting. Third, XGBoost allows users to define custom optimization goals and evaluation criteria, increasing its flexibility. Moreover, XGBoost implements parallel processing when selecting the best split node for enumeration, substantially improving the computational efficiency compared with Gradient Boosting Machine (GBM). The critical parameters of XGBoost are

n_estimators, learning rate, max_depth, and scale_pos_weight. The default value of scale_pos_weight is 1, indicating the positive and negative samples are in equilibrium. This is not applicable for precipitation classification in the cold season. More attention should be paid to scale_pos_weight when model training. The optimal parameters are shown in Appendix C.

### 3.2.4 MLR

The MLR is the first type of regression algorithm used extensively in many fields, assuming a stable linear relationship between a dependent variable and multiple independent variables. Compared with nonlinear relationships, the MLR is easier to fit and each explanatory variable's statistical property is more intuitive. MLR is usually fitted using the ordinary least square method to minimize the sum of squares of residuals predicted by the model and observed by the sample. The overall model for MLR is:

$$Y = \beta_0 + \beta_1 X_1 + \beta_2 X_2 + \cdots + \beta_i X_i, \quad i = 1, \dots, n, \tag{5}$$

Where $n$ is the number of explanatory variables, $Y$ is the dependent variable predicted by $X_1, X_2 \dots, X_n$. $\beta_0$ is the intercept, and $\beta_1, \beta_2 \dots, \beta_i$ are regression coefficients.

### 3.3 Performance evaluation and comparison

In this study, the performance of all products is evaluated using 692 randomly selected independent gauges from 2000 to 2017. The evaluation metrics mainly involve categorical and statistical metrics. The categorical metrics focus on analyzing the ability of products to capture precipitation events, including the probability of detection (POD), false alarm ratio (FAR), critical success index (CSI), Precision (*precision*), frequency bias (FB), Heidke Skill Score (HSS), and classification accuracy (*Accuracy*). The POD also called hit bias, represents the probability of precipitation events correctly detected. FAR and *precision* describe the ratio of falsely and correctly detected events among total detected precipitation events, respectively. The sum of FAR and *precision* is 1. The CSI incorporates POD and FAR, which demonstrates the overall ability of precipitation detection. The FB is the ratio of POD and FAR. It shows the balanced ability of products in detecting precipitation events. FB < 1 indicates that precipitation events are underestimated, and FB > 1 indicates overestimated. The FB equals 1 meaning that the number of missed events equals false alarmed events. HSS compares the predicted performance with random chance. The negative HSS shows random chance is better than the model predicted. The range of HSS is -∞ to 1, the perfect value is 1.

$$POD = \frac{H}{H+M}, \tag{6}$$

$$FAR = \frac{F}{H+F}, \tag{7}$$

$$Precision = \frac{H}{H+F}, \tag{8}$$

$$CSI = \frac{H}{H+M+F} \ , \tag{9}$$

$$FB = \frac{POD}{Precision} = \frac{H+F}{H+M} \ , \tag{10}$$

$$HSS = \frac{2(HN-FM)}{(H+M)\cdot(M+N)+(H+F)\cdot(F+N)} \ , \tag{11}$$

The *Accuracy* shows the proportion of total days that are correctly classified as wet and dry days. One point that needs to be emphasized is that this study takes *Accuracy* as the evaluation metric to describe the accuracy of ML classification models (RF, GBDT, and XGBoost) in training processes, thereby determining the optimal parameters of the model.

$$Accuracy = \frac{H+N}{H+M+F+N} \times 100\%, \tag{12}$$

Where $H$ is the total number of precipitation events simultaneously observed and predicted, $M$ is the total number of precipitation events observed but not predicted, $F$ is the total number of precipitation events predicted but not detected, $N$ is the total number of no-precipitation events. The optimal value of POD, *precision*, CSI, *Accuracy*, and FB is 1, while FAR is 0. The statistical metrics are used to evaluate the error of precipitation intensity, including root mean square error (RMSE), the modified Kling-Gupta efficiency (KGE) and its components (Pearson correlation coefficient (CC), bias ($\beta$) and variability ratio ($\gamma$)). The CC measures the magnitude of the correlation between the model predicted and observed values. The RMSE accesses the error between predicted and observed values. The KGE combining the CC, $\beta$, and $\gamma$ reflects the overall goodness of fit between model predicted and observed. $\beta > 1$ indicates precipitation amount is overestimated and vice versa. The formulas for these metrics are expressed as follows:

$$KGE = 1 - \sqrt{(CC-1)^2 + (\beta-1)^2 + (\gamma-1)^2} \tag{13}$$

$$CC = \frac{\sum_{i=1}^{n}(P_{oi}-\overline{P_o})(P_{mi}-\overline{P_m})}{\sqrt{\sum_{i=1}^{n}(P_{oi}-\overline{P_o})^2 \cdot (P_{mi}-\overline{P_m})^2}} \tag{14}$$

$$\beta = \frac{\mu_m}{\mu_o} \ , \tag{15}$$

$$\gamma = \frac{SD_m/\mu_m}{SD_o/\mu_o} \ , \tag{16}$$

$$RMSE = \sqrt{\frac{1}{n}\sum_{i=1}^{n}(P_{mi}-P_{oi})^2}, \tag{17}$$

Where $P_o$ and $P_m$ are the value of gauge observed and predicted precipitation, respectively. $N$ is the total number of samples. $\mu_m$ and $\mu_o$ are the mean value of gauge observed and predicted precipitation. $SD_o$ and $SD_m$ are the standard deviation of gauge observed and predicted precipitation, respectively. The optimal value for CC, KGE, $\beta$, $\gamma$ is 1, while for MAE and RMSE is 0.

# 4 Result

## 4.1 Evaluation the precipitation detection ability of MSMPs

The classification accuracy (*Accuracy*) of different ML models for wet/dry days is shown in Table 2. The general performances are considerable. The *Accuracy* for the three models is higher than 91% in the whole period, which is 91.8%, 91.7%, and 91.8% for RF, GBDT, and XGBoost, respectively. The *Accuracy* in the cold season is better than that in the warm season. There is no significant difference among the three classification algorithms. The main reason is that the input variables used in this study are sufficient in variety and quantity.

**Table 2. The classification accuracy (*Accuracy*) of wet/dry day during the warm season and cold season**

|              | RF   | GBDT | XGBoost |
|--------------|------|------|---------|
| Cold season  | 93.6 | 93.5 | 93.6    |
| Warm season  | 89.9 | 89.8 | 89.9    |
| Whole period | 91.8 | 91.7 | 91.8    |

To evaluate the efficiency of the proposed strategy in precipitation detection ability. The multi-source merged precipitation products (MSMPs: PGBDT2, PXGB2, and PRF2), gauge-based Kriging interpolated (Kriging), and original precipitation products (MSPs) are assessed and compared based on independent gauge observations. The six categorical metrics (POD, FAR, CSI, *precision*, FB, and HSS) are shown in Fig.5 and the average values of all gauges are expressed in Table 3. The overall accuracy of three MSMPs substantially outperforms other products. The best values of all metrics (except for POD) are generated in MSMPs. Kriging has the highest POD with a value of 0.93 (Fig. 5a), followed by ERA5L (0.94) and GSMaP (0.93). However, the POD of PGBDT2, PXGB2, and PRF2 are 0.84, 0.85, and 0.85, respectively. The FAR (Fig. 5b) of MSMPs is 0.13, decreased by 59 - 75% compared with the original MSPs (0.32-0.52). In addition, PRF2 obtains the highest CSI with a value of 0.76, much better than original MSPs (0.3-0.65) and Kriging (0.66) (Fig. 5c). In terms of *precision* (Fig. 5d), MSMPs show an obvious improvement. The *precision* increases from 0.48-0.68 (MSPs) to 0.87 (MSMPs). For FB (Fig. 5e), MSPs and Kriging deviate from 1, and PERCDR has the worst value (1.83). Although ERA5L achieves a high POD, its FB is 1.75, indicating ERA5L has seriously overestimated wet days and misclassified many precipitation events. Fortunately, MSMPs strike a good balance between hit and false alarmed rates. The FB of MSMPs is closer to 1, which is 0.96 for PGBDT2, 0.99 for PXGB2, and 0.98 for PRF2. In terms of HSS (Fig. 5f), except for Kriging (0.67) and GSMaP (0.66), the HSS of MSPs is lower than 0.5 (0.3-0.49). In contrast, the MSMPs (0.79-0.8) improve by 20 - 163%.

**Table 3. The average value of categorical metrics of multiple products compared with gauge observations during whole period.**

| Metrics | CHIRPS | CMORPH | PERCDR | GSMaP | IMERG | ERA5L | Kriging | PGBDT2 | PXGB2 | PRF2 |
|---------|--------|--------|--------|-------|-------|-------|---------|--------|-------|------|
| POD     | 0.36   | 0.70   | 0.75   | 0.93  | 0.78  | 0.94  | 0.95    | 0.84   | **0.85** | **0.85** |
| FAR     | 0.36   | 0.37   | 0.52   | 0.32  | 0.41  | 0.45  | 0.32    | **0.13** | **0.13** | **0.13** |

| | | | | | | | | | | |
|---|---|---|---|---|---|---|---|---|---|---|
| CSI | 0.30 | 0.48 | 0.39 | 0.65 | 0.50 | 0.54 | 0.66 | 0.75 | 0.75 | **0.76** |
| *precision* | 0.64 | 0.63 | 0.48 | 0.68 | 0.59 | 0.55 | 0.68 | **0.87** | **0.87** | 0.87 |
| FB | 0.61 | 1.20 | 1.83 | 1.39 | 1.38 | 1.75 | 1.45 | 0.96 | **0.99** | 0.98 |
| HSS | 0.30 | 0.48 | 0.31 | 0.66 | 0.49 | 0.49 | 0.67 | 0.79 | 0.79 | **0.80** |

Note: the values in bold are the best performing of each metric.

385  The general performance of most MSPs (e.g., CMORPH, PERCDR, and IMERG) in the warm season is better than that in the cold season (Fig. 5). However, the MSMPs' performance difference between warm and cold seasons is smaller than that of MSPs, demonstrating that the ability of MSMPs is more balanced throughout the year. Moreover, the metrics' variation of original MSPs is considerable in the cold season, particularly FAR and *precision*. The boxplot of FAR (Fig. 5b) and *precision* (Fig. 5d) for CHIRPS, CMORPH, and PERCDR have wider ranges, which represents these values are unevenly spatially

390  distributed. In contrast, MSMPs have more concentrated ranges of boxplots in most metrics. These results emphasize the necessity of prioritizing precipitation state recognition in the merging process, which can largely improve the precipitation capture efficiency of MSPs.

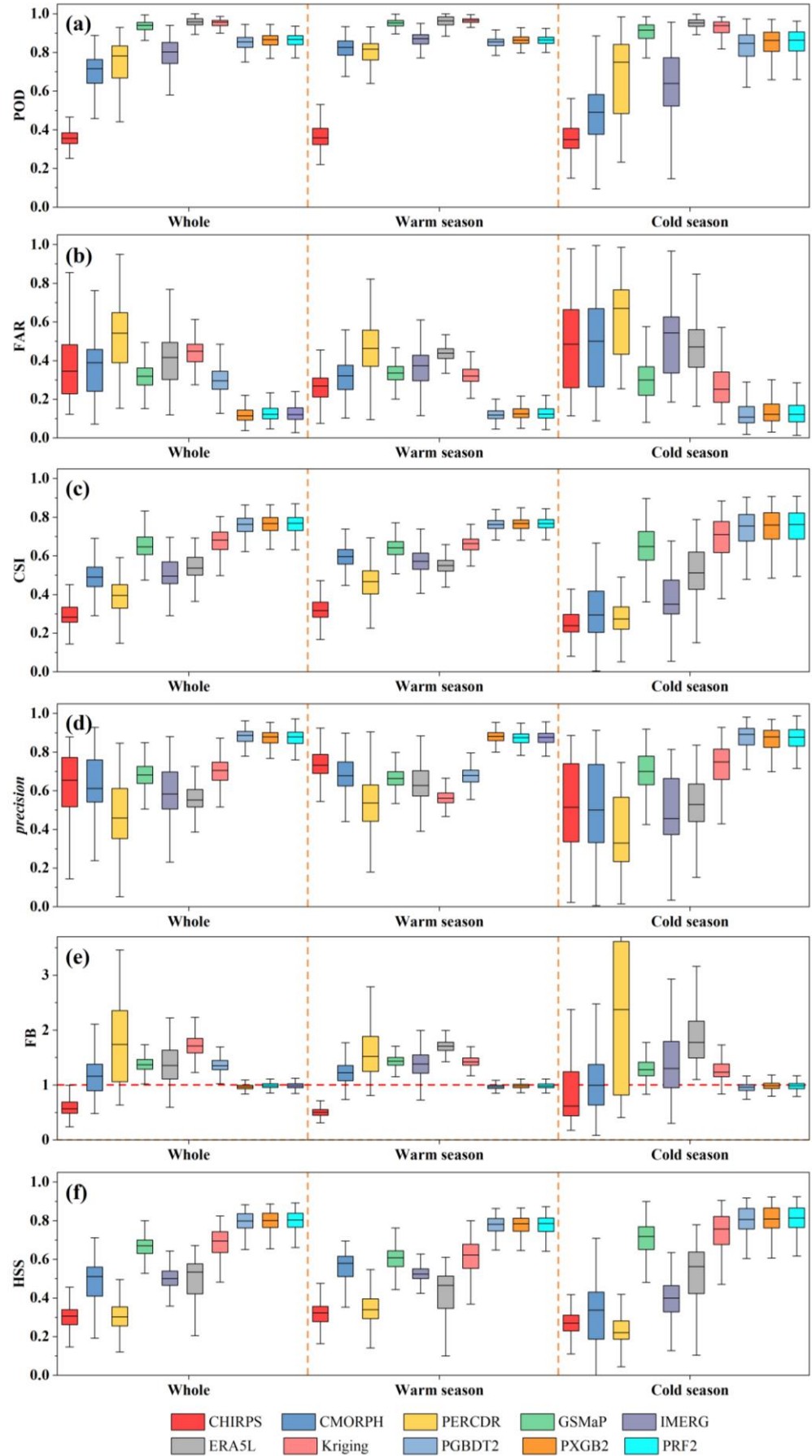

**Figure 5. Boxplots of six categorical metrics (POD, FAR, CSI, *precision*, FB, and HSS) for ten products, including six MSPs, one**

gauge-based interpolated data, and three ML-based merged data.

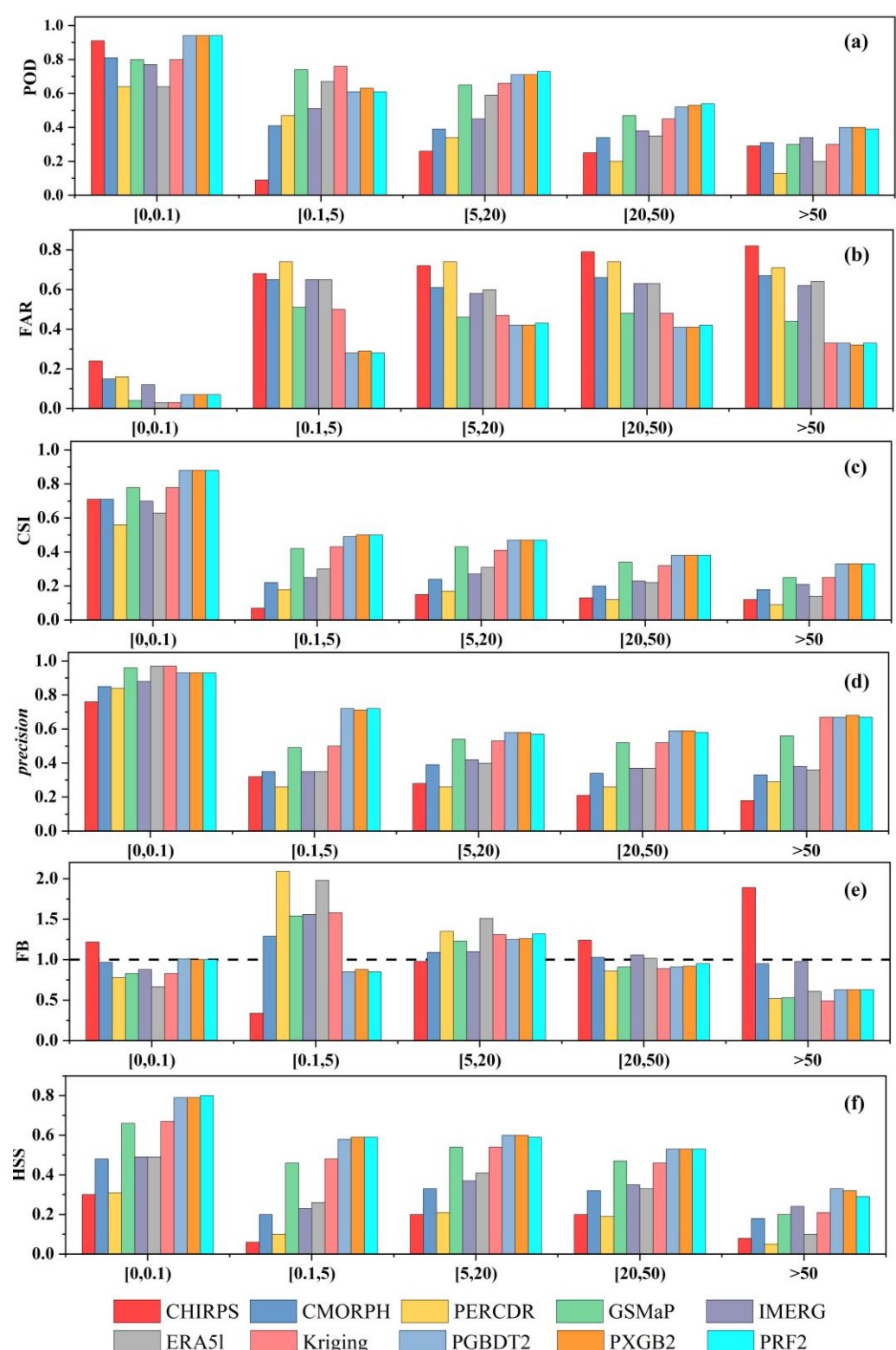

**Figure 6. The performance of six categorical metrics of ten products under various daily precipitation threshold.**

Fig. 6 shows the average value of six categorical metrics for ten products under different precipitation intensities, including no-precipitation (<0.1 mm/d), light precipitation ([0.1, 5)), moderate precipitation ([5, 20)), heavy precipitation ([20, 50)), and violent precipitation (>50 mm/d). Overall, MSMPs have the best performance regardless of precipitation intensities, followed by Kriging and GSMaP, signifying that ML classification techniques improve the detection capability of all precipitation thresholds, not only for light and moderate precipitation events. The performance of all products for no-

precipitation is considerably better than other precipitation intensities. For instance, the FAR, CSI, and HSS of MSMPs are 0.07, 0.88, and 0.79-0.8, respectively, in no-precipitation. Most MSPs have a poor ability to capture light and moderate precipitation (0.1-20 mm/d). MSPs' CSI range between 0.07-0.43 and HSS is 0.06-0.54, while the HSS of MSMPs varies between 0.58 to 0.6. In addition, the FB fluctuates greatly in light precipitation, with the lowest value of 0.34 for CHIRPS and the largest value of 2.09 for PERCDR (Fig. 6e). The MSMPs show the best FB values of 0.85. The accuracy begins to decrease when precipitation intensity is above 20mm/d (i.e., heavy and violent precipitation). For violent precipitation (> 50mm/d), the accuracy reduction of MSMPs and Kriging is relatively tiny compared with original MSPs. MSMPs have the highest POD (0.39-0.4), CSI (0.33), and HSS (0.47). However, the FAR and *precision* show a different trend with better accuracy in violent precipitation than in moderate and heavy precipitation (Fig. 6b, d). In addition, although the POD of ERA5L and Kriging outperform MSMPs in whole events, they are inferior to MSMPs in moderate, heavy, and violent precipitation. Generally, XGBoost and RF models are slightly superior to GBDT when dividing precipitation thresholds (Fig. 6a). Kriging exhibits better performance than most original MSPs. Nevertheless, it is only based on gauge observations and does not combine other climate variables associated with precipitation processes. When MSPs, gauge, and multiple covariates are considered, the MSMPs are more accurate than Kriging.

## 4.2 Evaluation the precipitation amounts of MSMPs

To explore the accuracy of precipitation amounts of MSMPs. Five statistical metrics (RMSE, KGE, and its components: CC, β, and γ) are employed to compare original MSPs and Kriging with PGBDT2, PXGB2, and PRF2 based on daily observations. According to comparison results (Fig. 7, Table 4), the MSMPs perform better than all original MSPs. The KGE of MSPs has been improved by 15-85% in the whole period (Fig. 7a). The KGE is 0.74-0.76 for MSMPs, 0.62 for Kriging, and 0.34-0.66 for MSPs. MSMPs have a strong correlation with gauge observations in the warm season (CC: 0.83), cold season (CC: 0.9), and the whole period (CC: 0.85) (Fig. 7b), which is substantially better than MSPs (warm:0.45-0.75; cold: 0.45-0.83; whole: 0.47-0.76). In addition, the β shows that all MSPs and Kriging overestimate precipitation amounts (Fig. 7c). This overestimation is more prominent in the cold season, with values ranging between 5%-38%. In contrast, MSMPs show significant improvements and obtain better skills in all seasons. Although GSMaP and CMORPH have better performance than PRF2 during the warm season and whole period, they suffer from a large magnitude of overestimation (Kriging: 6%, CMORPH:13%) in the cold season. In terms of γ, the average variability ratio of CHIRPS, CMORPH, and IMERG is more consistent with 1 than MSMPs (Fig. 7d). However, they show more discreteness, particularly for CHIRPS. In comparison, the distribution of MSMPs values is more compact. The results indicate that MSMPs can merge the complementary advantages of original data and reduce errors to a large extent, especially in the cold season. For RMSE (Fig. 7e), the values in the warm season are higher than that in the cold season. This is because precipitation is mainly concentrated in the warm season, and higher precipitation amounts often lead to larger RMSE. The RMSE for MSMPs decreases by 16 - 52% compared with original

MSPs (4.99 - 8.85mm/d). Among MSMPs, PXGB2 exhibits the smallest RMSE with a value of 4.2 mm/d.

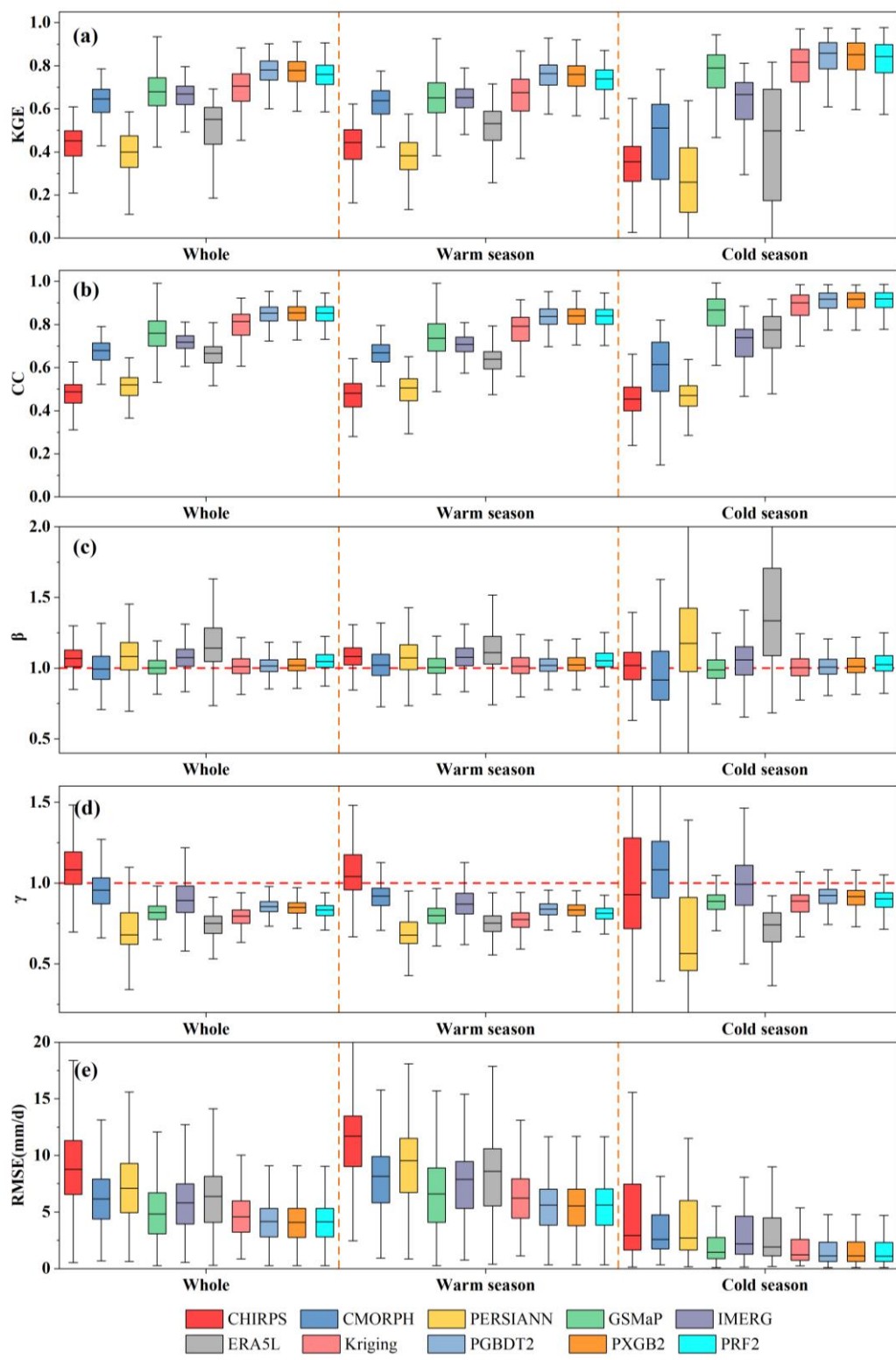

**Figure 7. Boxplots of five statistical metrics (CC, RMSE, KGE, β, and γ) for ten products.**

**Table 4. The average values of statistic metrices of multiple products compared with gauge observations during whole period (The unit of RMSE is mm/d).**

| Metrics | CHIRPS | CMORPH | PERCDR | GSMaP | IMERG | ERA5L | Kriging | PGBDT2 | PXGB2 | PRF2 |
|---|---|---|---|---|---|---|---|---|---|---|
| KGE | 0.41 | 0.58 | 0.34 | 0.66 | 0.64 | 0.48 | 0.62 | **0.76** | **0.76** | 0.74 |

| CC | 0.47 | 0.66 | 0.51 | 0.76 | 0.71 | 0.66 | 0.78 | **0.85** | **0.85** | **0.85** |
| β | 1.09 | 1.05 | 1.14 | 1.02 | 1.09 | 1.2 | 1.07 | **1.02** | 1.03 | 1.06 |
| γ | 1.1 | **0.95** | 0.71 | 0.82 | 0.9 | 0.74 | 0.78 | 0.85 | 0.84 | 0.83 |
| RMSE | 8.85 | 6.29 | 7.22 | 4.99 | 5.94 | 6.36 | 4.81 | 4.22 | **4.20** | 4.22 |

Note: the values in bold are the best performing of each metric.

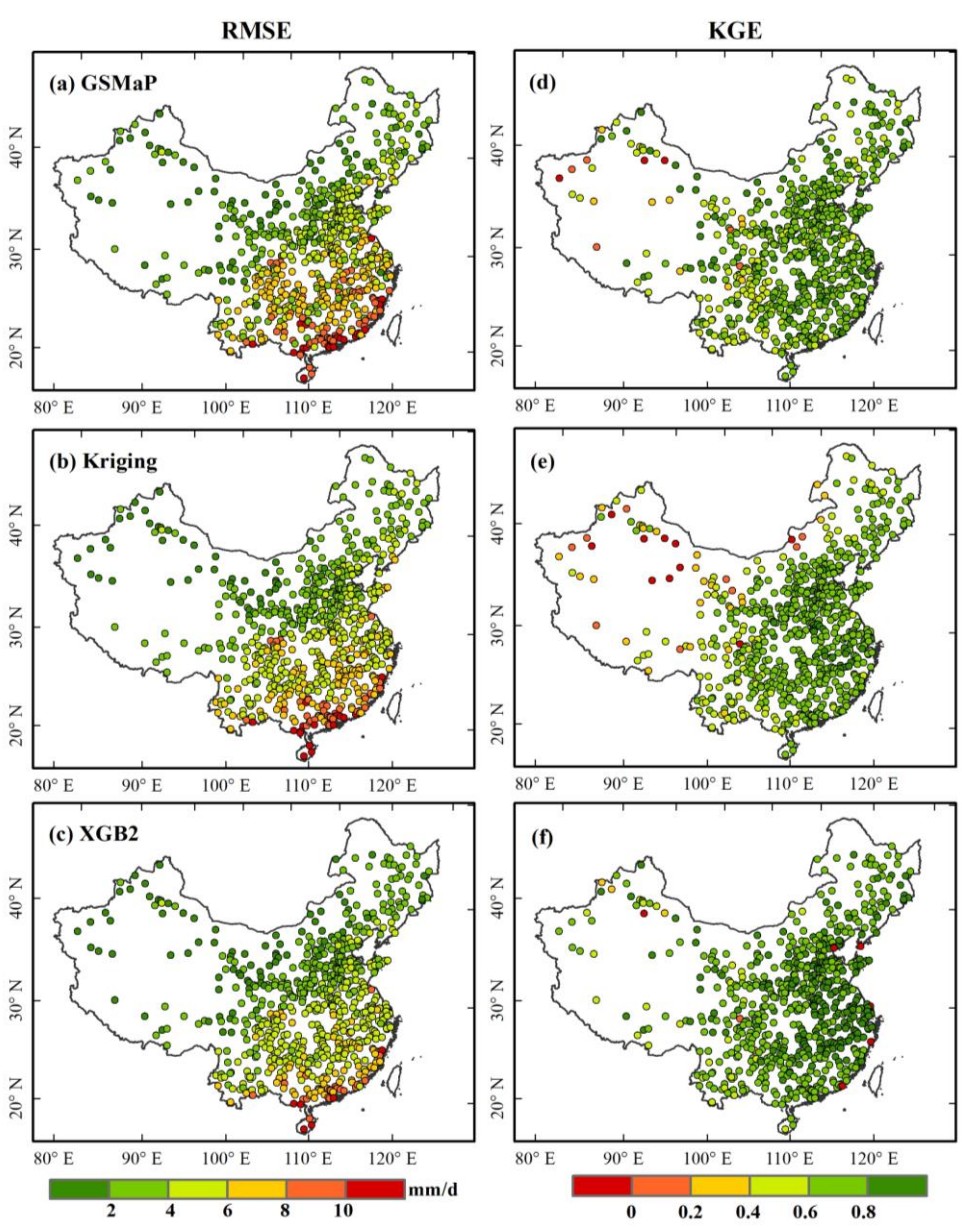

**Figure 8. Spatial distribution of RMSE (a-c) and KGE (d-e) for GSMaP (a, d), Kriging (b, e), and PXGB2 (c,f) in the whole period**

**from 2000-2017 using independent rain gauges over mainland China.**

Fig.8 illustrates the spatial distribution of RMSE and KGE for GSMaP, Kriging, and PXGB2 in the whole period. The

reason for showing only these three products is that they perform better among original products and MSMPs. The spatial

comparison among them is more representative and brevity. The RMSE gradually increases from north to south, which is

consistent with the precipitation change pattern (Fig. 8a). The PXGB2's RMSE in south China has better performance than

Kriging and GSMaP. For PXGB2, approximately 48% of the gauges have RMSE less than 4 mm/d. The percentage of gauges with RMSE higher than 8 mm/d is 14% for GSMaP, 8% for Kriging, and 4% for PXGB2. In addition, the spatial distribution of KGE shows that the low values are mainly gathered in the northwest (Fig. 8d-f). About 36% of the gauges with KGE higher than 0.8 for PXGB2, while only 15% for GSMaP and 30% for Kriging. The PXGB2 improves KGE performance over the northwest region and narrows the gap between the southeast and northwest regions. These results indicate that the two-step merging approach could mitigate the spatial variability of products and is less susceptible to topography.

**4.3 Variable importance of ML models**

The variable importance can quantitatively explain their contribution to improving model accuracy and recognize crucial input variables. The permutation feature importance is utilized to calculate variable importance values of models. The basic idea of this method is to randomly shuffle the order of a specific variable while keeping other variables unchanged and compute the accuracy difference (the evaluation metric is *Accuracy* for the classification model, mean squared error for the regression model) with the original model. As shown in Fig. 9, the importance of variables for GBDT, XGBoost, and RF and their ranks are different, which is related to the inherent structure of each model. This phenomenon also exists between classification and regression models. Nonetheless, KP is always the most important variable in each model, proving that the Kriging_based predictor considering the spatial autocorrelation between rain gauges is pretty helpful to improve model efficiency. For all models, the top three variables in importance are KP, GSMaP, and IMERG. The CMORPH, PERCDR, ERA5L, and temperature is considered next significant. The importance of ERA5L and temperature in XGBoost and RF classification models is more obvious than that in regression models. Additionally, longitude, latitude, DEM, cloud cover, and relative humidity exhibit relatively low influence on precipitation merging. The impacts of CHIRPS, soil moisture, and wind speed on prediction results are negligible. However, this does not mean that these predictors are not important for precipitation in whole regions. The slight importance of the latter variables may be affected by data quality and the correlation degree with precipitation. For example, CHIRPS is the worst performance product among original MSPs. Overall, it is necessary to employ multiple covariables in classification and regression models since complex precipitation processes cannot be thoroughly described by a single variable.

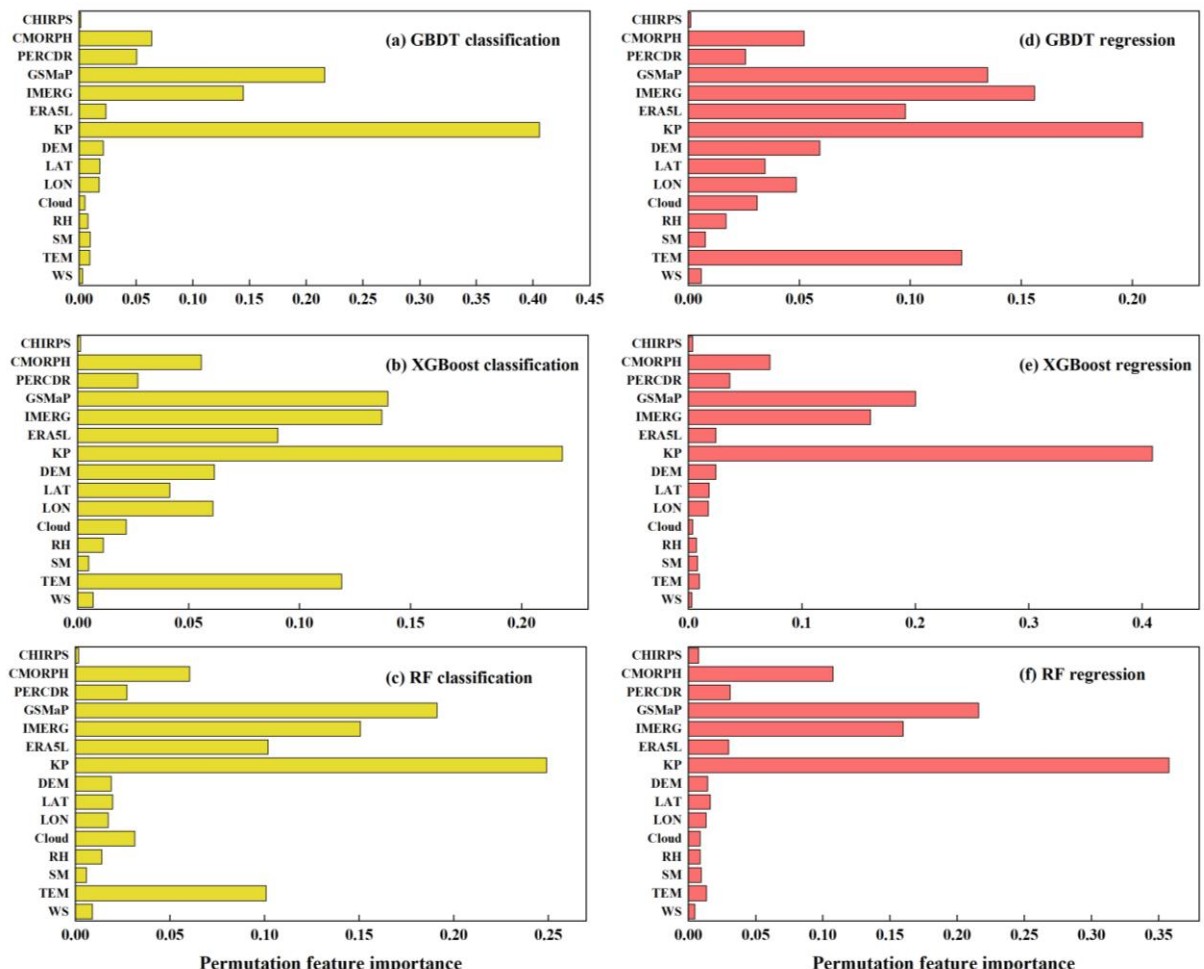

**Figure 9. Permutation feature importance of three (GBDT, XGBoost, and RF) classification models (a-c) and three regression model (d-f) in the warm season (LAT is latitude, LON is longitude, RH is relative humidity, SM is soil moisture, TEM is temperature, and WS is wind speed).**

## 5    Discussion

### 5.1 Comparison of the different merging strategies

From the aspect of merging processes, different models and training samples could affect the accuracy of the integrated dataset. Therefore, three additional merging scenarios are considered for quantitative comparison with the proposed strategy to highlight the impact of samples' division and algorithm selection on fusion results. Fig. 10 gives a brief overview of four scenarios and their corresponding merged precipitation products. Scenario1 is the method adopted in this study; scenario 2 separately trains model in each season based on four regression models (GBDT, XGBoost, RF, and MLR), the corresponding results are PGBDT_R, PXGB_R, PRF_R, and PMLR; scenario3 applies classification and regression models during the entire period, the results are PGBDT_E, PXGB_E, and PRF_E; while scenario4 solely employs four regression models during the entire period, the results are PGBDT_ER, PXGB_ER, PRF_ER, and PMLR_ER.

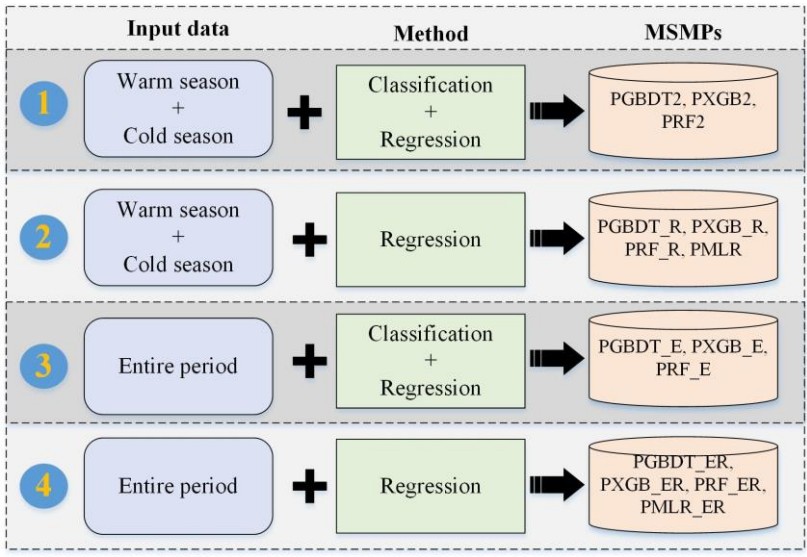

**Figure 10. Four scenarios with different sample periods and different models**

Fig. 11 shows the evaluation results (CC, CSI, KGE, FB, and HSS) of four scenarios between 14 MSMPs and independent gauge observations. The performance of scenario1 is apparently better than other scenarios. For scenarios2, although the statistical metrics (CC and KGE) are only slightly worse than scenario1, the categorical metrics (CSI, FB, and HSS) are considerably weakened. In the whole period (Fig. 11a), the HSS is between 0.64-0.68 for scenario2, much lower than 0.79-0.8 for scenario1. Moreover, the FB of scenario2 is larger than 1.38 (Fig. 11a), indicating that the number of precipitation events have been overestimated. A similar phenomenon also occurs in warm and cold seasons (Fig. 11b, c). Meanwhile, the MLR performs worse than the three ML models. The results of scenario2 demonstrate that only relying on regression models to merge precipitation can describe precipitation intensity but not capture precipitation occurrence well. In terms of scenario3, the overall performance is superior to scenario2 but inferior to scenario1. The CSI (Fig. 11c) for scenario1 and scenario3 range from 0.73-0.74 and 0.70-0.72, respectively. Scenario3 suggests that merging precipitation in different seasons could balance the performance differences within a year. Scenario4 shows the worst performance regardless of season, with poor CSI, FB, and HSS. Especially for the PMLR-ER dataset, its accuracy is even worse than GSMaP and Kriging. This is because MLR is difficult to describe the complex relationship between precipitation and other variables. The four scenarios can be ranked by prediction accuracy from best to worst: Scenario1 > Scenario3 > Scenario2 > Scenario4. The approach (i.e., Scenario1) employed in this study is proved to be more accurate than other traditional strategies.

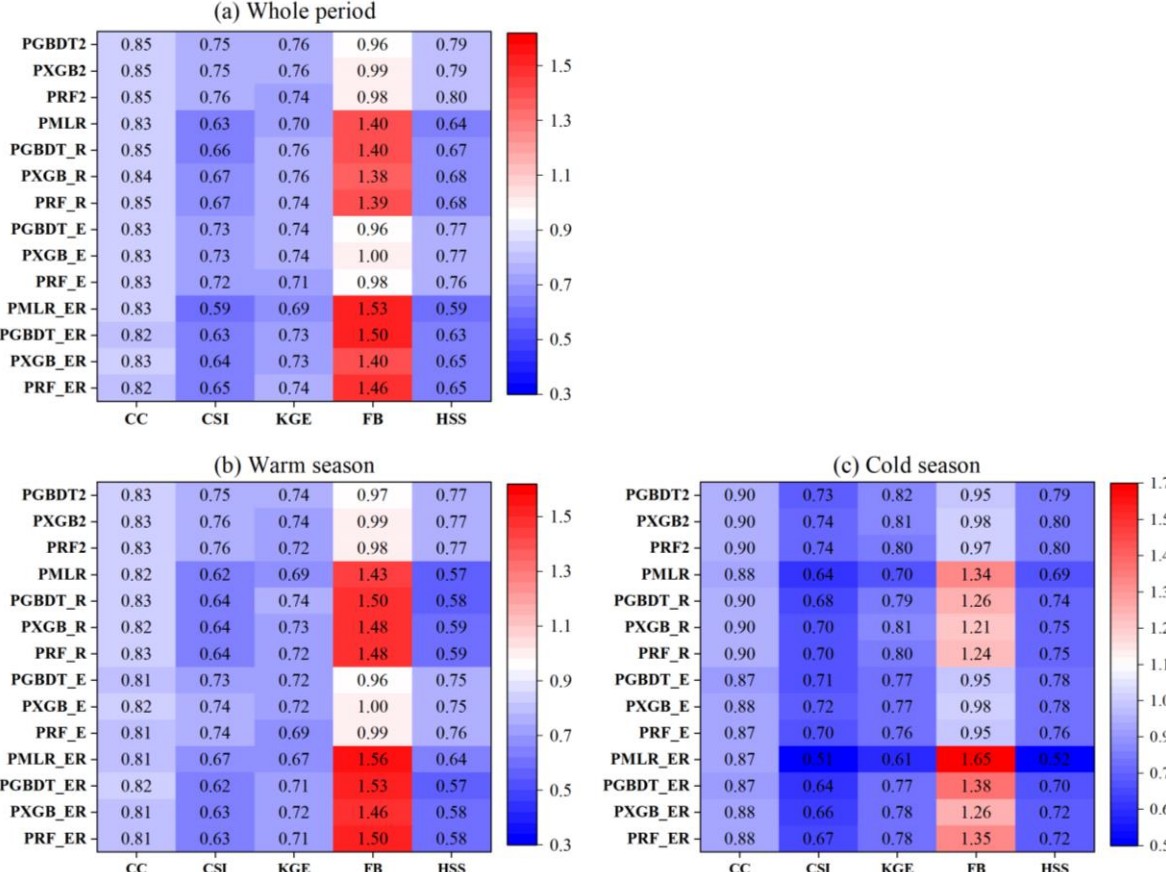

**Figure 11. Five evaluation metrics (CC, CSI, KGE, FB, and HSS) for different products under four scenarios during the whole period, warm season, and cold season.**

## 5.2 Models efficiency

The GBDT, XGBoost, and RF models show similar improvements in the two-step merging strategy. Nevertheless, different models have their inherent advantages and disadvantages. There is an apparent disproportion between positive and negative samples (wet and dry days) when training the classification model, which directly impacts the model's classification accuracy. In this study, the proportion of positive and negative samples in the cold season is approximately 1: 3.2. In terms of this imbalance problem, RF and XGBoost algorithms have built-in parameters to adjust. However, GBDT requires additional oversampling methods such as the Synthetic Minority Over-sampling Technique (SMOTE) method to solve, which increases the complexity of model training. Meanwhile, it can be inferred from the results of Table 3, Fig. 5, and Fig. 11 that the FB of XGBoost outperforms RF in all seasons, indicating XGBoost has better equilibrium ability for disproportional samples. In addition, Fig. 12 displays the computational costs of training for three models under different sample sizes. The result exhibits that the training time of GBDT and RF is much higher than XGBoost, which is mainly related to the model structure and parallel training. XGBoost parallels the feature granularity rather than the tree granularity. The most time-consuming part of decision tree learning is sorting feature values to determine the optimal split node. XGBoost ranks the values before training and then saves them into a block structure, which is repeatedly used in subsequent iterations. In this way, the training time

could be vastly reduced (Chen et al., 2016; Wang et al., 2019). Therefore, considering the complexity, accuracy, and computational costs of the model, XGBoost is an optimal choice for predicting daily precipitation over China.

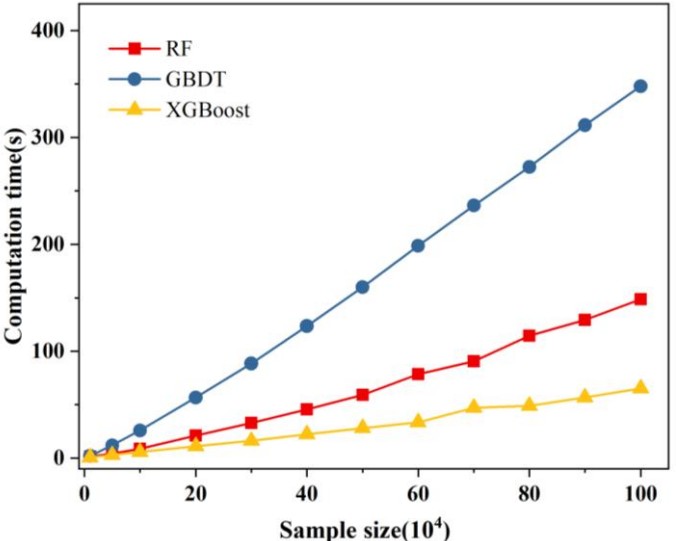

**Figure 12. Comparison of computation time of three ML classification models.**

## 5.3 The influence of gauge density and spatial resolution

The density of rain gauges could influence the performance of the merged product as well as the gauge-based interpolated product. Gauges with different densities are used to train model and interpolation, including 10%, 30%, 50%, and 70% of total gauges. Fig. 13 shows the higher gauge density leads to the better performance of the merged and interpolated products. However, PXGB2 is less affected by the density than Kriging. The decreased magnitude of Kriging's accuracy is more significant than PXGB2's as the gauge number is reduced. For instance, the deterioration of the KGE is 0.04 for PXGB2 (0.76 to 0.72) but 0.32 for Kriging (0.63 to 0.31), which is also smaller than Baez-Villanueva et al. (2020) and Zhang et al. (2021). The precipitation capture efficiency of PXGB2 decreases slightly and always shows a better performance. The CSI and HSS of PXGB2 vary from 0.73-0.76 and 0.77-0.79, respectively. The FB is relatively stable under different gauge numbers. In addition, even gauge density is reduced to 10% (237 gauges, i.e., 40,000 km$^2$ per gauge), PXGB2 also outperforms Kriging at 70% (1680 gauges) and the best original MSPs (i.e., GSMaP). In comparison, the performance of Kriging is inferior to GSMsP when gauge density is less than 50%, especially at 10%, which shows the gauge-based interpolation method is more suitable for gauge density regions and could lead to considerable uncertainties in low gauge density regions. In general, these results demonstrate that the proposed method is effective and robust, and it is expected to be applied to improve precipitation accuracy in areas with scarce data.

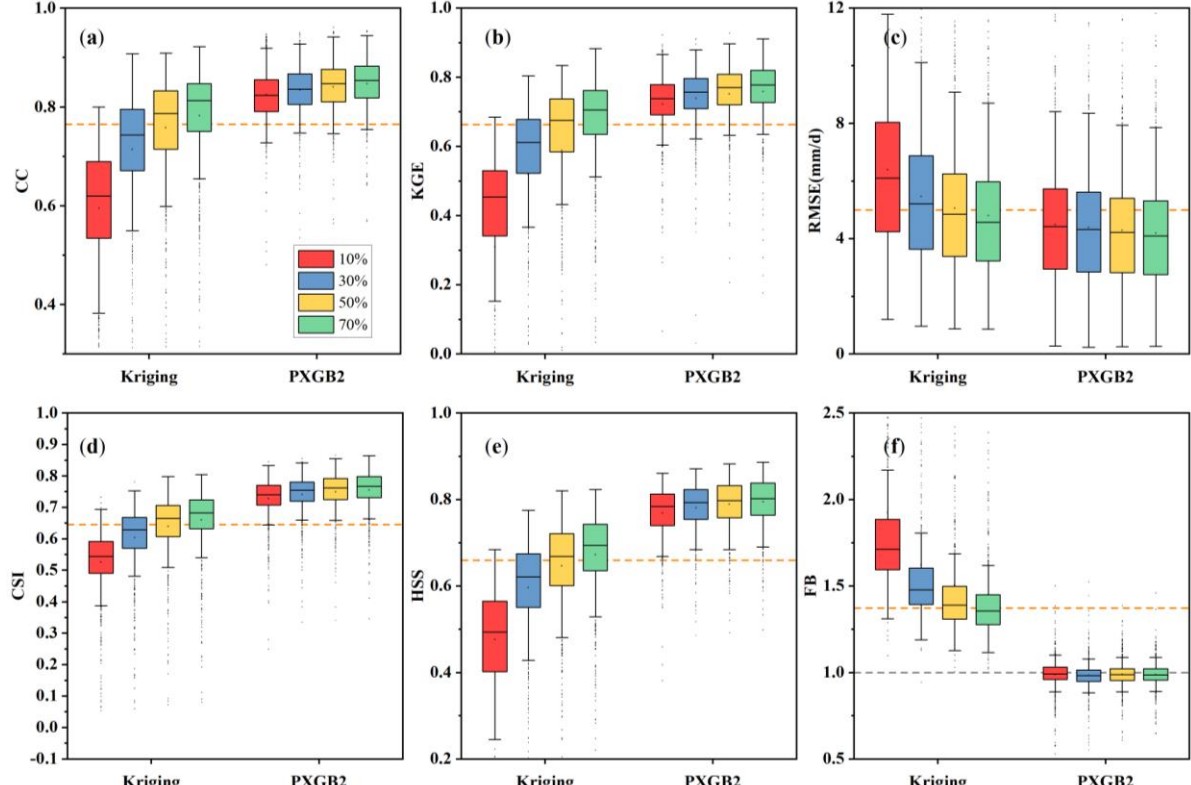

**Fig. 13 Performance of PXGB2 and Kriging products using training dataset with different rain gauge densities (10%, 30%, 50, 70%). The dotted orange line shows the average of the best original product (GSMaP). The gray dotted line in (f) represents the reference line with a value of 1.**

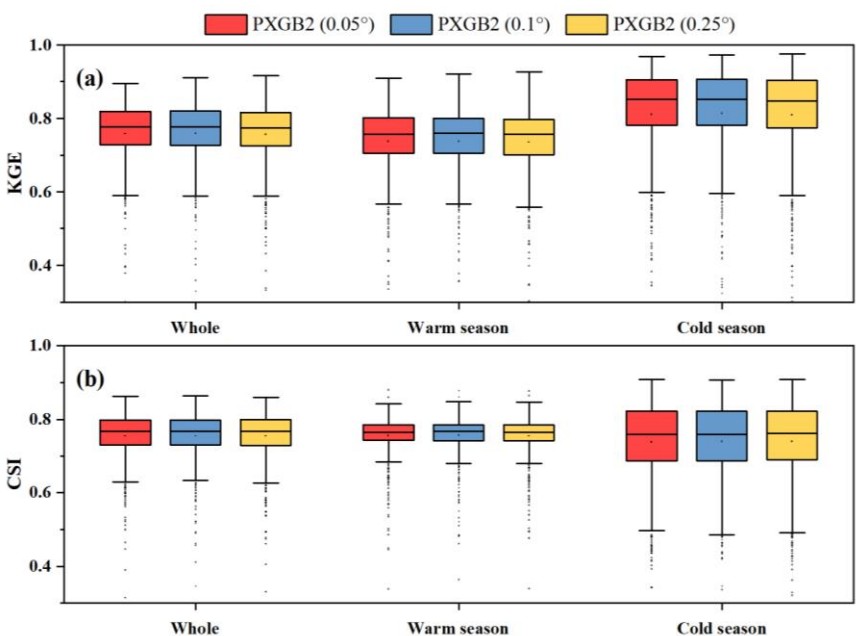

**Fig. 14 The performance (KGE and CSI) of PXGB2 prepared MSPs with different spatial resolutions (0.05°, 0.1°, and 0.25°) during the whole period, warm season, and cold season.**

This study uses a simple interpolation method to resample products to keep a consistent spatial resolution and avoid additional uncertainties, as many previous studies have done (Chao et al., 2018; Zhang et al., 2020; Baez-Villanueva et

al., 2020; Wu et al., 2020; Wang et al., 2020; Hong et al., 2021). Fig. 14 shows the performance of PXGB2 obtained by training models with precipitation products under different spatial resolutions (0.05°, 0.1°, and 0.25°). It demonstrates that there are only slight differences between various resolutions during the whole period as well as warm and cold seasons, which is consistent with the previous study (Baez-Villanueva et al., 2020). Therefore, it can be considered that unifying the spatial resolution of all products to 0.1° has a negligible impact on the merging results in this study.

**5.4 Comparison with the previous studies**

The study combines classification and regression models to improve the accuracy of MSPs, which pays special attention to optimizing precipitation detection ability and reducing the error caused by missed and false alarms. This research has made significant progress based on the achievements that previous studies have been done. In terms of precipitation occurrence, the classification accuracy (91.8%) is better than the ANN model (86.5%) applied by Xiao et 560 al. (2020) and the RF model (77.5%) employed by Pham et al. (2019). The POD of MSMPs is lower than GSMaP and ERA5L, which is similar to Xiao et al. (2020). In addition, Yin et al. (2021) improved the CC of the original product by 11% and RMSE by 7% over China, which is slightly inferior to the improvement of this study (CC and RMSE improved 12% and 16%, respectively). Furthermore, the overall performance of MSMPs is substantially better and could provide more accurate precipitation information for hydrological research. The CC of MSMPs is up to 0.85, much higher than 565 0.78 in Zhang et al. (2021), 0.61 in Yin et al. (2021), and 0.72 in Wu et al. (2020) over China. Although the validation method and period vary in different studies, their conclusions still have reference value. The outperformance of this study is mainly due to the consideration of precipitation products from multiple sources, environmental variables, and relatively higher gauge density. Most importantly, the spatial autocorrelation considered in this study plays an important role in the merging process. Compared with considering spatial distance (Baez-Villanueva et al. (2020), geographical 570 coordinates, and spatial correlation (Zhang et al., 2021), it can not only describe spatial autocorrelation between gauges but also between rain gauges and predicted points. In addition, some previous studies based on statistical methods were complex and difficult to reproduce for researchers in other fields (Yang et al., 2017; Ma et al., 2021; Yin et al., 2021). For instance, Yang et al. (2017) combined the MSPs and gauges by bias correction, gauge observation gridding, and data merging. In comparison, the proposed method only relies on ML and does not involve other statistical methods, which 575 is easy to implement and has strong transferability.

**5.5 Limitation and uncertainties**

Although this proposed merging strategy has achieved outstanding performance, some issues still need to be discussed and further improved in future studies. The gauge observations are taken as the reference in model training and evaluation. However, it suffers from uncertainties induced by diverse climates, complex topography, and measuring instruments (Ma et

al., 2015; Lei et al., 2021). These uncertainties are more obvious in the gauges located in snow and glacier coverage regions and would be propagated to merged precipitation results. Moreover, gauges at high altitudes are sparsely distributed and have strong spatial heterogeneity, making it challenging to describe precipitation distribution accurately. In future studies, the input datasets could be divided into more groups according to different terrain or altitude zones, and precipitation data in high altitude regions could be corrected by combining topographic factors, snowfall, and glacier mass balance data to mitigate their uncertainties.

This study assumes that the rain gauge represents the areal precipitation pattern in its corresponding grid, but this assumption is not fully satisfied in practical application, especially in the Tibetan Plateau. This spatial scale mismatch problem between precipitation gridded and single gauge observations could be alleviated by downscaling coarse products to a finer resolution. Some studies have downscaled all products before merging them with gauge observations (Chen et al., 2018; Chen et al., 2021). However, downscaling daily precipitation is challenging because it is difficult to describe the relationship between precipitation and environmental variables (Chen et al., 2021). More effective downscaling algorithms are worth exploring in the future.

Due to the limitation of gauge observations, the benchmark and MSPs used in this study are not near-real-time products. The merged products are more suitable for studying hydrometeorological changes in long time series than in the middle or short term. Multi-source precipitation products with near real-time and finer temporal resolution can be continuously merged, such as IMERG Early Run and GSMaP_NRT, to improve the accuracy of precipitation for flood prediction if rain gauges are available. In addition, although the trained model has spatial transferability, there is uncertainty when applied to precipitation prediction outside the training period.

## 6 Conclusion

This study proposes a two-step merging strategy including GBDT, XGBoost, and RF classification and regression algorithms to merge multi-sources precipitation products, multiple environment variables, and rain gauges from 2000 to 2017 over China. The performance of three merged products (MSMPs) is validated based on 692 randomly selected independent gauges and compared with original MSP, Kriging, and other traditional merging scenarios (e.g., ML regression and MLR). Several statistical and categorical metrics are employed to quantitatively describe the precipitation detection capability and precipitation uncertainties. The main findings of this study can be concluded as follows:

(1) The precipitation capture ability of MSPs has been substantially improved. The MSMPs are better than all original MSPs and Kriging regardless of the precipitation intensity. The CSI for MSPs and Kriging is 0.30-0.65 and 0.66, while MSMPs are increased to 0.75-0.76. The HSS has also been improved by 21-166 % (0.79-0.8) compared with MSPs (0.30-0.66).

(2) The statistical biases of precipitation amounts induced by hit events are obviously alleviated. The improvement of CC, KGE, and RMSE is 12-81%, 15-85%, and 16-52%, respectively. The spatial difference in precipitation accuracy between northwest and southeast China is also narrowed.

    (3) It is essential to incorporate spatial autocorrelation in the merging strategy. Kriging-based predictor (KP) is the most important covariable in precipitation merging, followed by GSMaP, IMERG, and ERA5L. The degree of importance

for covariables in models also relates to their inherent accuracy.

    (4) Compared with traditional MLR and ML regression models, the proposed method in this study has superior performance in all aspects. Meanwhile, the MSMPs predicted by considering annual precipitation characteristic distribution are better than those in the whole period.

    (5) The higher gauge density used in model training could lead to a better performance of the proposed method. However,

this method could also remarkably improve original products even with few gauges.

    (6) The comprehensive ability of RF and XGBoost is slightly better than GBDT. Considering the computation efficiency, it is more recommended to use XGBoost to prepare merged precipitation products.

The two-step merging strategy proposed in this study achieves satisfactory performance over China. It is robust and efficient in such a region characterized by complex terrain, variable climate, and uneven distribution of gauges. Therefore, this

method has great referential significance and can also achieve excellent results when applied in other regions and countries.

*Data availability:* The rain gauge observations are obtained from the China Meteorological Data Service Center (http://data.cma.cn). The IMERG is from https://gpm1.gesdisc.eosdis.nasa.gov/data/GPM_L3/GPM_3IMERGDF.06/. The GSMaP is from http://sharaku.eorc.jaxa.jp/GSMaP/index.htm. The CHIRPS is from https://data.chc.ucsb.edu/products/CHIRPS-2.0/. The PERCDR is from https://www.ncei.noaa.gov/data/precipitation-

persiann/access/, The CMORPH is from https://ftp.cpc.ncep.noaa.gov/precip/CMORPH_V1.0/CRT/. The ERA5-Land is from https://cds.climate.copernicus.eu/. The GLDAS_NOAH is from https://disc.gsfc.nasa.gov/.

*Author contribution:* Huajin Lei designed the methodology and collected datasets, Huajin Lei and Hongyu Zhao implemented algorithms' code, Huajin Lei analyzed the results and wort the original manuscript. Tianqi Ao supervision. All the authors revised and improved the manuscript.

*Competing interests.* The contact author has declared that neither they nor their co-authors have any competing interests.

*Financial support.* The research is financially supported by the Science and Technology Foundation of Sichuan Province (no.2020FYQ0013).

**Appendix A:   The number and location of stations used in GPCC over China**

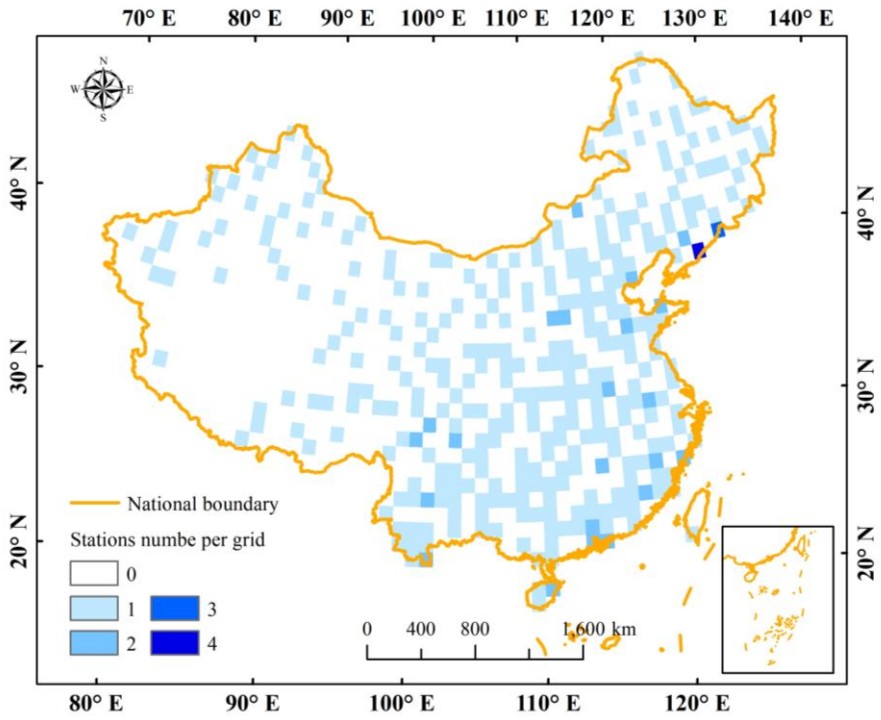

                        **Fig. S1 The number and location of stations used in GPCC over China**

From the latest GPCC dataset, the number of China's International Exchange Stations used in GPCC has fluctuated between

360-370 (In Fig. S1, the number is 362 July 2015), which has increased in recent years. Before 2017, only about 200 China's

stations are used in GPCC. Despite the use of these stations, satellite precipitation products are corrected based on monthly

GPCC, making it insufficient to improve daily performance.

**Appendix B:   Comparison of different semivariogram models**

The widely used semivariogram models include: spherical, exponential, Gaussian, power, and linear. We have discussed

the different of the Kriging_based prediction (KP) based on five semivariogram models. The expresses of five models

as follows:

(1)  Spherical model:

$$\gamma(h) = \begin{cases} 0 & h = 0 \\ C_0 + C\left(\frac{3}{2} \cdot \frac{b}{a} - \frac{1}{2} \cdot \frac{b^3}{a^3}\right) & 0 < h \le a \\ C_0 + C & h > a \end{cases} \tag{S1}$$

(2)  Exponential model:

$$\gamma(h) = \begin{cases} 0 & h = 0 \\ C_0 + C\left(1 - \exp\left(\frac{-h}{r}\right)\right) & h > 0 \end{cases} \tag{S2}$$

where $\gamma(h)$ is semivariogram, $h$ is the distance, $C_0$, $C$, and $a$ is the nugget, sill, and range, respectively.

(3)  Gaussian model:

$$\gamma(h) = \begin{cases} 0 & h = 0 \\ C_0 + C\left(1 - \exp\left(\frac{h^2}{r^2}\right)\right) & h > 0 \end{cases} \qquad \text{(S3)}$$

where the range is $\sqrt[2]{3}a$

(4) Power model:

$$\gamma(h) = h^a \qquad 0 < a \leq 2 \qquad \text{(S4)}$$

(5) Linear model:

$$\gamma(h) = \begin{cases} 0 & h = 0 \\ C_0 + C\left(\frac{h}{a}\right) & 0 < h \leq a \\ C_0 + C & h > a \end{cases} \qquad \text{(S5)}$$

In order to compared the performance of the five semivariogram models, the Kriging_based predictions (KP) of total 2372 gauges are estimated and validated. The accuracy of KP will directly influence the model training and merging results. The evaluated results of different model are show in Table R1.

Table S1 The performance of KPs estimated from five models

| Metrics | Spherical | Exponential | Gaussian | Power | Linear |
|---|---|---|---|---|---|
| CC | 0.806 | **0.810** | 0.782 | 0.799 | 0.803 |
| RMSE | 4.530 | **4.486** | 4.862 | 4.625 | 4.582 |
| RB | 0.028 | 0.032 | 0.044 | 0.040 | **0.006** |
| FAR | 0.276 | 0.284 | **0.269** | 0.302 | 0.282 |
| POD | 0.931 | **0.943** | 0.895 | 0.942 | 0.937 |
| CSI | **0.688** | 0.687 | 0.674 | 0.670 | 0.685 |
| KGE | **0.692** | 0.685 | 0.684 | 0.661 | 0.675 |
| $\beta$ | 1.028 | 1.032 | 1.044 | 1.040 | **1.006** |
| $\gamma$ | 0.830 | 0.816 | **0.876** | 0.798 | 0.814 |
| *precision* | **0.724** | 0.716 | 0.731 | 0.698 | 0.718 |
| HSS | **0.708** | 0.706 | 0.696 | 0.686 | 0.705 |

Note: the values in bold represent the best performing values.

It can be seen from Table R1 that the overall performance of five models is good. The performance of spherical model shows the best CC, RMSE, and RB. The exponential model shows the best CSI, KGE, *precision*, HSS. The difference of semivariogram models is relatively small and the spherical model with slight better performance is adopted in this study.


## Appendix C:    Model parameters

Table S2 The optimal parameters of RF model training

|  | Period | n_estimators | max_depth | min_samples_split |
|---|---|---|---|---|
| classification | warm | 150 | 60 | 7 |
|  | cold | 150 | default | 7 |
| regression | warm | 200 | default | 10 |
|  | cold | 200 | 70 | 4 |

Table S3 The optimal parameters of GBDT model training

|  | Period | n_estimators | max_depth | learning_rate |
|---|---|---|---|---|
| classification | warm | 100 | 9 | 0.2 |
|  | cold | 100 | 7 | 0.4 |
| regression | warm | 100 | 10 | 0.1 |
|  | cold | 200 | 9 | 0.1 |


Table S4 The optimal parameters of XGBoost model training

|  | Period | n_estimators | max_depth | learning_rate | scale_pos_weight |
|---|---|---|---|---|---|
| classification | warm | 100 | 10 | 0.2 | 1.1 |
|  | cold | 150 | 10 | 0.2 | 1.2 |
| regression | warm | 300 | 10 | 0.05 | 1 |
|  | cold | 150 | 9 | 0.1 | 1 |

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
