# Peer review of "A two-step merging strategy for incorporating multi-source precipitation products and gauge observations using machine learning classification and regression over China"

_Hydrology and Earth System Sciences, 2021_

## Author Comment (AC1)

Overall comments:

Thank you for providing me the opportunity to review the entitled manuscript " A two-step merging strategy for incorporating multi-source precipitation products and gauge observations using machine learning classification and regression over China". It's an interesting study and fits the scope of the Journal. However, there are several major and minor flaws that authors should take care of and revise before final consideration in high-quality peer-review Journals like Hydrology and Earth System Science.   Authors should improve the quality of the manuscript including research outcomes, discussion, and unique conclusions.

Response: Thanks very much for your valuable and meaningful suggestions on our manuscript. These comments significantly improve the quality of this manuscript. We have tried our best to carefully study the suggestions you raised and made corresponding modifications to the manuscript. The grammar of the manuscript has been polished by a native English speaker. The responses to the reviewer's comments are as follows:

1- In the abstract, there are too many simple conclusive statements in the abstract, which are well-known to people who are involved in this area. You should first layout a background information description. Second, point out what is the most important in the current field of research and what has not yet been solved. Third, explain your novel method in detail how you solve this problem. Last, a brief description of your own findings should be presented. If your methods and findings are novel and interesting to people who are involved in this area, they will continue to read your main text.

Response: the abstract has been rewritten to highlight the key information of this study according to your so patient suggestions. The revised Abstract is as follows:

**Abstract.** Although many multi-source precipitation products (MSPs) with high spatio-temporal resolution have been extensively used in water cycle research, they are still subject to various biases, including false alarm and missed bias. Precipitation merging technology is an effective means to alleviate this uncertainty. However, how to efficiently improve precipitation detection efficiency and precipitation intensity simultaneously is a problem worth exploring. This study presents a two-step merging strategy based on machine learning (ML) algorithms, including gradient boosting decision tree (GBDT), extreme gradient boosting (XGBoost), and random forest (RF). It

incorporates six state-of-the-art MSPs (GSMaP, IMERG, PERSIANN-CDR, CMORPH, CHIRPS, and ERA5-Land) and rain gauges to improve the accuracy of precipitation from precipitation identification and estimation during 2000-2017 over China. Multiple environment variables and spatial autocorrelation are combined in the merging process. The strategy first employs classification models to identify wet and dry days and then combines regression models to predict precipitation amounts based on classified wet days. The merged results are compared with traditional methods, including multiple linear regression (MLR), ML regression models, and gauge-based Kriging interpolation. A total of 1680 (70%) rain gauges are randomly chosen for model training and 692 (30%) for performance evaluation. The results show that: (1) The multi-sources merged precipitation products (MSMPs) outperformed all original MSPs in terms of statistical and categorical metrics, which substantially alleviates the bias and deviation in temporal and spatial. The modified Kling-Gupta efficiency (KGE), critical success index (CSI), and Heidke Skill Score (HSS) of original MSPs have been improved by 15-85%, 17-155%, and 21-166%, respectively. (2) The spatial autocorrelation plays a significant role in precipitation merging, which considerably improves the model accuracy. (3) The performance of MSMPs obtained by this method is superior to MLR, Kriging interpolation, and ML regression models. XGBoost algorithm is more recommended for large-scale data merging owing to its high computational efficiency. (4) The two-step merging strategy performs better when higher density gauges are used to model training. But it has strong robustness and can also obtain better performance than original MSPs even when the gauges number is reduced to 10%. This study provides an accurate and reliable method to improve precipitation accuracy under complex climatic and topographic conditions. It could be applied to other areas well if rain gauges are available.

2- It is recommended that the author rewrite the INTRODUCTION, increase the citation of the literature, and extract questions and useful information from the literature. Through literature review, point out the shortcomings of existing research, thus leading to the article's hydrological and environmental significance and purpose. In this section, the literature review needs to be more critical.

Response: Thanks! Your suggestions are very helpful in improving the quality of introduction. Through reviewing more literatures, the shortcomings of existing research and advantage of this

research has been concluded. The Introduction has been rewritten in the revised manuscript.

3- The authors should detail the methodological novelties with the vast amount of existing literature in this area in the Introduction.

Response: The methodological novelties have been elaborated in detail in the Introduction. The corresponding contents have been modified in the revised manuscript.

4- The authors should address the clear objectives of the study in the introduction section.

Response: The objectives of the study have been rewritten to make it more clear for readers and are as follow:

The objectives of this study mainly include three-folds: The objectives of this study mainly include three-folds: (1) exploring the effectiveness of the proposed strategy in all aspects according to various metrices; (2) comparing the performance of the proposed strategy with traditional methods; (3) assessing the influence of MSPs' spatial resolution and gauge density on model performance. This strategy is expected to improve the accuracy of existing MSP and explore the potential of more ML algorithms in precipitation.

5- In the discussions, comparisons of the results obtained in this manuscript with the extensive existing literature on Satellite-based precipitation and the methodologies used need to be expanded. I recommend authors should compare results with previous approaches. In discussions, it must add what the results mean with respect to what is already known and highlight how your results support or refute the current hypotheses in the field if any. More references should be added to that section. Underline how your results make a significant move in the working field forward.

Response: Thanks for your suggestions, which greatly improved the quality of the Discussion. The method proposed in this study have been compared with extensive previous studies from the aspects of implement difficulty, efficiency, and accuracy. An individual section is added in Discussion to detailly state the new information of the results and highlight the superiority of the proposed method. The detailed information has been added in the revised manuscript.

6- It's is recommended to improve the quality of grammar and take care of grammar mistakes.

Response: the grammar problems in the whole manuscript have been carefully checked and

improved by a native English speaker.

7- Line# 115-120 Expand the hydro-metrological features of the study region with more explanations (Study area).

Response: The description of the hydro-metrological features of the study region have been expanded in the revised manuscript.

8- Line# 230-240 I am wondering if the technique applied by authors is correctly classified for wet and dry events. Authors only attempted to correct precipitation events fall in wet events. I recommend techniques should test with combined dry and wet days because measurement techniques for all precipitation datasets are quite different and definitely there would be a lag between wet and dry events for all different datasets which could create outliers in applied techniques.

Response: The biases of precipitation products mainly come from overestimating/underestimating the amounts of hit events, and failing to correctly distinguish precipitation occurrence, including false alarm and missed events. It is difficult to reduce all biases by directly correcting the precipitation amount of all samples. This is the main reason why we chose a two-step methodology to merge products.

The classification model determines whether a day is a wet day or a dry day according to the classification probability. In this way, all days are classified into wet or dry days. The classification results are evaluated with 692 independently rain gauges, and the evaluated result show classification models show better performance. In addition, as you said, the techniques should test with combined dry and wet days. I am not sure I have correctly understood your suggestion, i.e., directly using the regression models to merge precipitation products including dry and wet days of the entire events rather than distinguishing dry and wet days separately (Fig. R1). To compare the difference between the two techniques, the merged results of using only regression models and using both classification and regression models are shown in discussion, the main conclusions are as follows (Table R1):

[Figure]

Fig. R1 two merging techniques

Table R1 The compared results of the two techniques

|  | MSMPs | CC | CSI | KGE | FB | HSS |
|---|---|---|---|---|---|---|
| Method 1 | PGBDT2 | 0.85 | 0.75 | 0.76 | 0.96 | 0.79 |
|  | PXGB2 | 0.85 | 0.75 | 0.76 | 0.99 | 0.79 |
|  | PRF2 | 0.85 | 0.76 | 0.74 | 0.98 | 0.80 |
| Method 2 | PMLR | 0.83 | 0.63 | 0.70 | 1.40 | 0.64 |
|  | PGBDT_R | 0.85 | 0.66 | 0.76 | 1.40 | 0.67 |
|  | PXGB_R | 0.84 | 0.67 | 0.76 | 1.38 | 0.68 |
|  | PRF_R | 0.85 | 0.67 | 0.74 | 1.39 | 0.68 |

It can be seen from Table R1 that the technique (Method 1) applied in this study is obviously better that the technique combined the wet and dry days based on regression models. The performance of categorical metrics (CSI, FB, and HSS) is higher and the statistical metrics (CC and KGE) is also well. Therefore, it is reliable and beneficial to improve the accuracy of merged products to firstly classify wet and dry events.

9- Authors have merged ground observations data with different precipitation estimates. While different precipitation products have different spatial resolutions which could cause outliers when merging with ground observations. Apart from these, the authors used a simplified approach to merge coarse resolution precipitation products with ground observations data. Rainfall within a single satellite pixel could vary considerably by 38% between two gauges located within 4 km × 4 km. The deviation between gridded precipitation and single gauge observation is due to the discrepancy of scale and could be reduced by increasing the validation stations or downscaling the TRMM precipitation to a finer resolution. Therefore, authors should firstly downscale all grided

precipitation datasets at a finer scale and then merge with ground observational data to make the approach more novel towards environmental significance. The author can learn lessons from the following papers.

Response: The spatial scale mismatch between gauge and pixel has always been a great challenge in satellite products verification and fusion. We assumed that rain gauges represent the areal precipitation in their corresponding pixels. It's not easy to downscale the daily precipitation with long periods (2000-2017) to a finer resolution (1km or 5km) over such a large area (about 9.6 million km$^2$). Several precipitation products used in this study. It would be time-consuming and lead to massive amount of data to downscale all of them, which will further bring a computational difficulty to next data fusion grid by grid. Meanwhile, most studies focused to downscale precipitation at monthly or annual scales (Jia et al., 2011; Shi et al., 2015; Jing et al., 2016; Ma et al., 2017; Chen et al., 2018; Chen et al., 2021; Ghorbanpour et al., 2021; Shen and Yong, 2021), the land surface variable with high resolution is unavailable at the daily scale, such as the 8d Land surface temperature product (MOD10A2) and 16d Normalized vegetation index products (MOD13Q1). Therefore, downscaling precipitation at higher temporal scales (e.g., daily or hourly) is challenging since the relationships between environmental variables and precipitation on these scales are far less evident and difficult to capture (Chen et al., 2021).

In addition, this study focuses on high-efficiency precipitation merging methods rather than downscaling methods. As many previous studies have done (Chao et al., 2018; Zhang et al., 2020; Baez-Villanueva et al., 2020; Wu et al.,2020; Wang et al., 2020; Hong et al., 2021), we use simple interpolation methods to resample product to keep its value and avoid other additional uncertainties. Meanwhile, considering the IMERG is the extension of TMPA, ERA5 has already superseded ERA-Interim (discontinued from August 2019). For reproducibility purposes, we disregarded TMPA and replaced ERA-Interim with ERA5-Land product. Therefore, six products are used in this study, among which four products (IMERG, GSMaP, CHIRPS, and ERA5-Land) have a spatial resolution of less than or equal to 0.1°, and only two products (CMORPH and PERSIANN-CDR) have a coarse spatial resolution of 0.25°. We resampled the latter products into 0.1° to keep the same resolution with other products, and believe that have a slight impact on the merging results. To prove this statement, we discuss the effect of using products with different spatial resolutions in Discussion.

In the future, many efforts will be focused on the downscaling method based on this study to

obtain high precision-resolution daily precipitation products over China.

Jia, S., Zhu, W., Lǚ, A., Yan, T., 2011: A statistical spatial downscaling algorithm of TRMM precipitation based on NDVI and DEM in the Qaidam Basin of China. Remote sensing of Environment, 115(12), 3069-3079.

Shi, Y., Song, L., Xia, Z., Lin, Y., Myneni, R. B., Choi, S., Wang, L., Ni, X., Lao, C., Yang, F., 2015. Mapping annual precipitation across mainland China in the period 2001–2010 from TRMM3B43 product using spatial downscaling approach. Remote Sensing, 7(5), 5849-5878.

Jing, W., Yang, Y., Yue, X., & Zhao, X., 2016: A spatial downscaling algorithm for satellite-based precipitation over the Tibetan plateau based on NDVI, DEM, and land surface temperature. Remote Sensing, 8(8), 655.

Ma, Z., Shi, Z., Zhou, Y., Xu, J., Yu, W., & Yang, Y., 2017: A spatial data mining algorithm for downscaling TMPA 3B43 V7 data over the Qinghai–Tibet Plateau with the effects of systematic anomalies removed. Remote Sensing of Environment, 200, 378-395.

Chen, Y., Huang, J., Sheng, S., Mansaray, L. R., Liu, Z., Wu, H., Wang, X., 2018: A new downscaling-integration framework for high-resolution monthly precipitation estimates: Combining rain gauge observations, satellite-derived precipitation data and geographical ancillary data. Remote Sensing of Environment, 214, 154-172.

Ghorbanpour, A. K., Hessels, T., Moghim, S., Afshar, A., 2021. Comparison and assessment of spatial downscaling methods for enhancing the accuracy of satellite-based precipitation over Lake Urmia Basin. Journal of Hydrology, 596, 126055.

Shen, Z., Yong, B., 2021. Downscaling the GPM-based satellite precipitation retrievals using gradient boosting decision tree approach over Mainland China. Journal of Hydrology, 602, 126803.

Baez-Villanueva, O.M., Zambrano-Bigiarini, M., Beck, H.E., McNamara, I., Ribbe, L., Nauditt, A., Birkel, C., Verbist, K., Giraldo-Osorio, J.D., Xuan Thinh, N.: RF-MEP: A novel Random Forest method for merging gridded precipitation products and ground-based measurements, Remote Sens. Environ., 239, 111606, https://doi.org/10.1016/j.rse.2019.111606, 2020

Chen, C., Hu, B., Li, Y.: Easy-to-use spatial Random Forest-based downscaling-calibration method for producing high resolution and accurate precipitation data, Hydrol. Earth Syst. Sci., https://doi.org/10.5194/hess-2021-332, 2021.

Chao, L., Zhang, K., Li, Z., Zhu, Y., Wang, J., Yu, Z.: Geographically weighted regression based methods for merging satellite and gauge precipitation, J. Hydrol., 558, 275-289, https://doi.org/10.1016/j.jhydrol.2018.01.042, 2018

Zhang, L., Li, X., Cao, Y., Nan, Z., Wang, W., Ge, Y., Wang, P., Yu, W.: Evaluation and integration of the top-down and bottom-up satellite precipitation products over mainland China. J. Hydrol. 581, 124456, 2020.

Wu, H., Yang, Q., Liu, J., Wang, G.: A spatiotemporal deep fusion model for merging satellite and gauge precipitation in China, J. Hydrol., 584, 124664, https://doi.org/10.1016/j.jhydrol.2020.124664, 2020.

Wang, Y., Wang, L., Li, X., Zhou, J., Hu, Z.: An integration of gauge, satellite, and reanalysis precipitation datasets for the largest river basin of the Tibetan Plateau. Earth System Science Data, 12(3), 1789-1803, 2020.

Hong, Z., Han, Z., Li, X., Long, D., Wang, J.: Generation of an improved precipitation data set from multisource information over the Tibetan plateau. J. Hydrometeorol., https://doi.org/10.1175/JHM-D-20-0252.1, 2021.

10- The study region covers complex hydro-topographical features and some of the stations are located in snow and glacier coverage regions (e.g., Tibetan Plateau) and hence observed precipitation in these regions is unreliable and unavailable. Therefore, the orographic correction of precipitation based on the vertical gradients along with glacier mass balance is required to retrieve an accurate precipitation dataset in high-altitude mountain regions such as Tibetan Plateau and some others. Authors can take glacier mass variations from GRACE data and try to correct precipitation for high-altitude regions.

Response: Thank you very much for your useful and novel suggestions.

The gauge observations suffer from some uncertainties, including wind-induced undercatch, wetting loss, and evaporation loss, which are more obvious in mountainous areas with complex terrain. Nevertheless, we take gauge observations as the true value in many literatures because it is the most direct means of obtaining precipitation (Lei et al., 2021; Xiao et al., 2022; Xu et al., 2022). In general, the gauges are used as the benchmark to validate and correct the errors of satellite products.

There are few attempts to use satellite products to correct observed precipitation, because additional errors may be caused due to the inherent uncertainties in products. Most importantly, even correcting for observation precipitation with glacier mass balance, it is difficult to assess the reliability of corrected observations due to the lack of validation data.

The glacier mass balance from GRACE has uncertainty (Velicogna and Wahr, 2013; Wouters et al., 2019; Wang et al., 2020) and is also suffered from some limitations. This uncertainty will be further propagated to precipitation correction. GRACE has a shorter temporal span and is discontinuous in time. Meanwhile, it has a coarse temporal resolution (monthly) and a spatial resolution (0.5° or 1°), while the temporal/spatial resolution of this study is daily/0.1° from 2000-2017. Using GRACE for precipitation correction may lead to spatio-temporal scale mismatch and may not achieve the desired effect. In addition, the application of GRACE in Tibetan Plateau is different because it is strongly influenced by land water storage, such as the increase of water storage in lakes on the Tibetan Plateau (Wang et al., 2020). Therefore, it is a promising and challenging work to use glacier mass balance data for precipitation correction at high latitudes.

This issue is a limitation of this study and explained in the Discussion in the revised manuscript. In further work, we will consider the Tibetan Plateau separately and add more auxiliary data (such as snow depth and snow water equivalent) to further optimize precipitation at high altitudes.

Lei, H., Li, H., Zhao, H., Ao, T., Li, X.: Comprehensive evaluation of satellite and reanalysis precipitation products over the eastern Tibetan plateau characterized by a high diversity of topographies, Atmos. Res., 259, https://doi.org/10.1016/j.atmosres.2021.105661, 2021.

Xiao, S., Zou, L., Xia, J.: Bias correction framework for satellite precipitation products using a rain/no rain discriminative model, Sci. Total Environ., https://doi.org/10.1016/j.scitotenv.2021.151679, 2021.

Xu, J., Ma, Z., Yan, S., Peng, J.: Do ERA5 and ERA5-land precipitation estimates outperform satellite-based precipitation products? A comprehensive comparison between state-of-the-art model-based and satellite-based precipitation products over mainland China. Journal of Hydrology, 605, 127353.2022.

Velicogna, I., Wahr, J.: Time-variable gravity observations of ice sheet mass balance: Precision and limitations of the GRACE satellite data. Geophysical Research Letters, 40(12), 3055-3063, 2013.

Wouters, B., Gardner, A. S., Moholdt, G.: Global glacier mass loss during the GRACE satellite mission (2002-2016). Frontiers in earth science, 7, 96, 2019.

Wang, Q., Yi, S., & Sun, W.: Continuous estimates of glacier mass balance in high mountain Asia based on ICESat-1, 2 and GRACE/GRACE follow-on data. Geophysical Research Letters, 48(2), e2020GL090954, 2021

11- Add a limitation section that explains any limitations that your hypothesis or experimental approach might have and the reasoning behind it and some of them I have clearly mentioned. This will help the field in generating hypotheses and new approaches without facing the same challenges. The discussion becomes well-rounded when you emphasize not only the impact of the study but also where possibly it falls short. Consider posing a few questions or directions, preferably in the form of a hypothesis, to provide a launchpad for future research.

Response: the limitations and uncertainties of this study have been added in Discussion. It explains the uncertainties of gauge observations, spatial scale mismatch between precipitation products and gauges, and other limitations about this study. Meanwhile, it also puts forward some directions for future efforts and improvement. The detained contents are added in the revised manuscript.

12- Conclusions need to revise and well-round major and significant findings of current study

Response: the conclusions have been improved in this revised manuscript.

---

## Author Comment (AC2)

Overall comments:

The article titled "A two-step merging strategy for incorporating multi-source precipitation products and gauge observations using machine learning classification and regression over China" assesses the effectiveness of three machine learning-based algorithms to merge precipitation products over China. The article is very well written, concise, relevant, and I enjoyed reading it. I believe that it fulfills the requirements to be published in HESS. In the following points, I describe some points and suggestions that I think can be considered to increase the quality of the manuscript.

Response: Thanks very much for your valuable and meaningful suggestions on our manuscript. These comments significantly improve the quality of this manuscript. We have tried our best to carefully study the suggestions you raised and made corresponding modifications to the manuscript. The grammar of the manuscript has been polished by native English speakers. The responses to the reviewer's comments are as follows:

1- ERA5 has already superseded ERA-Interim; therefore, I recommend using ERA5 instead. Similarly, the authors used TMPA 3B42 (with IMERG), which is no longer in production. For reproducibility purposes, it is essential to use products that can still be acquired.

Response: This is a nice suggestion considering future product updates. As you said, ERA5 has already superseded ERA-Interim and IMERG superseded TMPA. IMERG inherits the advantages of TRMM and makes many improvements. Therefore, we remove TRMM and keep IMERG. The previous study (Xu et al., 2022) demonstrated that the overall performance of ERA5-Land is superior to ERA5 at the daily scale, the ERA5-Land is used to replace ERA-Interim. Hence, six precipitation products are used in this study, including IMERG, GSMaP, CHIRPS, CMORPH, PERSIANN-CDR, and ERA5-Land.

Xu, J., Ma, Z., Yan, S., Peng, J., 2022: Do ERA5 and ERA5-land precipitation estimates outperform satellite-based precipitation products? A comprehensive comparison between state-of-the-art model-based and satellite-based precipitation products over mainland China. Journal of Hydrology, 605, 127353.

2- I suggest including a figure showing the number and location of stations used in the GPCC product in the appendix or supplement material. This information will be beneficial for the readers.

Response: The figure showing the number and location of stations used in the GPCC product have

been added in the revised manuscript in the appendix, and shown as follows:

[Figure]

Fig. S1 The number and location of stations used in GPCC over China

From the latest GPCC dataset, the number of China's International Exchange Stations used in GPCC has fluctuated between 360-370 (In Fig. S1, the number is 362 July 2015), which has increased in recent years. Before 2017, only about 200 China's stations are used in GPCC (Tang et al., 2016). Despite the use of these stations, satellite precipitation products are corrected based on monthly GPCC, making it insufficient to improve daily performance.

Tang, G., Ma, Y., Long, D., Zhong, L., Hong, Y., 2016: Evaluation of GPM Day-1 IMERG and TMPA Version-7 legacy products over Mainland China at multiple spatiotemporal scales, Journal of hydrology, 533, 152-167.

3- It would be helpful to describe whether the ground-based data were quality controlled.

Response: The ground-based data were quality controlled and the corresponding description are added in the revised manuscript.

4- The readers would benefit substantially from an improved description of the methodology. I liked Section 3.1 as it is very clear and informative, as well as the description of the machine learning algorithms and their respective figures. However, I believe that the description of the two-step merging strategy (Section 3.2) can be further improved. The authors mention that first, the gauge

observations are classified into wet and dry days to be used as the benchmark for classification models. Can the authors explain what the reason behind this is? It is not clear how the dry days were separated from wet days. How is a day classified as dry? Is the classification performed by grid-cell and by day? What is the final result of this classification? How can data scarcity affect this classification? Solving these questions is crucial for the understanding of the manuscript. Additionally, the authors mention that a regression process was applied to the wet days for the cold and warm periods. How were the models trained independently for the warm and cold periods? Perhaps it would be helpful to do a diagram for the first and second steps where the process is clearly explained. This will increase the manuscript's impact and is essential as the article presents these novel merging procedures.

Response: Thank you for your encouragement and suggestions. Your suggestions and questions are answered point by point as follows:

(1) Q: The authors mention that first, the gauge observations are classified into wet and dry days to be used as the benchmark for classification models. Can the authors explain what the reason behind this is?

A: The biases of precipitation products mainly come from overestimating/underestimating the amounts of hit events, and failing to correctly distinguish precipitation occurrence, including false alarm and missed events. It is difficult to reduce all biases by directly correcting the precipitation amount of all samples. Correctly judging whether precipitation occurs is an important way to improve precipitation detection efficiency. Therefore, the purpose of the first step is to classify precipitation to reduce the missed and false alarmed bias.

(2) Q: It is not clear how the dry days were separated from wet days. (a) How is a day classified as dry? (b) Is the classification performed by grid-cell and by day? (c) What is the final result of this classification?

A: **(a)** The gauge observations are distinguished to wet/dry days according to the 0.1mm/d threshold value (Lei, et al., 2020; Yu et al., 22020; Jiang et al., 2021) and used as the benchmark for classification, the wet day is set as 1, dry day is set as 0. The feature values of MSPs and covariables corresponding to each grid are applied to construct XGBoost, GBDT, and RF classification models, respectively. The model determines whether a day in the grid is a wet day or a dry day according to the classification probability, i.e., if the probability of a day >0.5, it is

judged as a wet day and the output value is 1. Otherwise, it is judged as a dry day and the return value is 0. **(b)** The classification is performed by grid-cell in time series. The orange dotted line in Fig. R1 shows the process of model applying grid by gird. **(c)** Hence, the classification result contains only wet and dry days (0,1) of each grid and does not involve precipitation intensity.

[Figure]

Fig. R1 the flowchart of classification

(3) How can data scarcity affect this classification?

The influence of data density (10%, 30%, 50%,70% of the total gauges) on classification result has been discussed in discussion part.

(4) Additionally, the authors mention that a regression process was applied to the wet days for the cold and warm periods. How were the models trained independently for the warm and cold periods? Perhaps it would be helpful to do a diagram for the first and second steps where the process is clearly explained.

According to the annual distribution characteristics of precipitation, we group all input datasets into two seasons: warm season (May and October) and cold season (November to April). The model is constructed and trained in warm and cold seasons using divided independent datasets, which leads to six classification and six regression models (i.e., two seasons with three models for classification and regression). Fig. 2 in the manuscript has illustrated the overall flowchart of merging strategy including the first and second steps, I hope that my detailed description can make reviewer and readers clearly understand the method of this study. The description of the methodology has also improved in the revised manuscript.

5- I liked the idea of using the semivariogram as a spatial autocorrelation variable :). Can the authors discuss the influence that the selection of a particular semivariogram model can have in applying

the method?

Response: Thank you for your suggestions. The widely used semivariogram models include: spherical, exponential, Gaussian, power, and linear. We have discussed the different of the Kriging_based prediction (KP) based on five semivariogram models. The expresses of five models as follows:

(1) Spherical model:

$$\gamma(h) = \begin{cases} 0 & h = 0 \\ C_0 + C\left(\frac{3}{2}\cdot\frac{b}{a} - \frac{1}{2}\cdot\frac{b^3}{a^3}\right) & 0 < h \le a \\ C_0 + C & h > a \end{cases} \qquad \text{(S1)}$$

(2) Exponential model:

$$\gamma(h) = \begin{cases} 0 & h = 0 \\ C_0 + C\left(1 - \exp\left(\frac{-h}{r}\right)\right) & h > 0 \end{cases} \qquad \text{(S2)}$$

where $\gamma(h)$ is semivariogram, $h$ is the distance, $C_0$, $C$, and $a$ is the nugget, sill, and range, respectively.

(3) Gaussian model:

$$\gamma(h) = \begin{cases} 0 & h = 0 \\ C_0 + C\left(1 - \exp\left(\frac{h^2}{r^2}\right)\right) & h > 0 \end{cases} \qquad \text{(S3)}$$

where the range is $\sqrt[2]{3}a$

(4) Power model:

$$\gamma(h) = h^a \qquad 0 < a \le 2 \qquad \text{(S4)}$$

(5) Linear model:

$$\gamma(h) = \begin{cases} 0 & h = 0 \\ C_0 + C\left(\frac{h}{a}\right) & 0 < h \le a \\ C_0 + C & h > a \end{cases} \qquad \text{(S5)}$$

In order to compared the performance of the five semivariogram models, the Kriging_based predictions (KP) of total 2372 gauges are estimated and validated. The accuracy of KP will directly influence the model training and merging results. The evaluated results of different model are show in Table R1.

Table R1 The performance of KPs estimated from five models

| Metrics | Spherical | Exponential | Gaussian | Power | Linear |
|---|---|---|---|---|---|
| CC | 0.806 | **0.810** | 0.782 | 0.799 | 0.803 |
| RMSE | 4.530 | **4.486** | 4.862 | 4.625 | 4.582 |

| | | | | | |
|---|---|---|---|---|---|
| RB | 0.028 | 0.032 | 0.044 | 0.040 | **0.006** |
| FAR | 0.276 | 0.284 | **0.269** | 0.302 | 0.282 |
| POD | 0.931 | **0.943** | 0.895 | 0.942 | 0.937 |
| CSI | **0.688** | 0.687 | 0.674 | 0.670 | 0.685 |
| KGE | **0.692** | 0.685 | 0.684 | 0.661 | 0.675 |
| $\beta$ | 1.028 | 1.032 | 1.044 | 1.040 | **1.006** |
| $\gamma$ | 0.830 | 0.816 | **0.876** | 0.798 | 0.814 |
| *precision* | **0.724** | 0.716 | 0.731 | 0.698 | 0.718 |
| HSS | **0.708** | 0.706 | 0.696 | 0.686 | 0.705 |

Note: the values in bold represent the best performing values.

It can be seen from Table R1 that the overall performance of five models is good. The performance of spherical model shows the best CC, RMSE, and RB. The exponential model shows the best CSI, KGE, *precision*, HSS. Therefore, there is slightly different between spherical and exponential models. While the latter three models (Gussian, Power, and Linear) are inferior to the former two. KP is an important variable in precipitation merging, its performance will directly affect the training accuracy of the ML model. However, the difference of semivariogram models is relatively small and the spherical model with slight better performance is adopted in this study. Therefore, the influence of different models on KP can reflect its influence in the merging method application.

6- For reproducibility purposes, please mention the parameters of the RF, GBDT, and XGBoost that were used while training the models during the warm and cold period.

Response: The optimal parameters of the RF, GBDT, and XGBoost classification and regression models during the warm and cold periods are displayed in the appendix.

7- The authors evaluated the performance of two categorical indices (CSI and HSS) over different precipitation intensities. This is a very good idea as the detection of no-precipitation events mainly masks the categorical performance. In this sense, as the objective of this article is to assess and compare the effectiveness of merging precipitation products using different ML techniques, I

suggest evaluating all categorical metrics over these precipitation intensities. This separation into rain intensities will provide additional insights regarding the performance of these merged products.

Response: Thank you for your suggestion. We have added all categorical metrics (POD, FAR, CSI, precision, FB, and HSS) over these precipitation intensities in the revised manuscript.

8- The first two sections of the Results are a bit puzzling for me. In Section 4.1, "Performance assessment for classification results", the results obtained during the first step of the merging procedure are shown. However, in this section, the authors evaluate precipitation intensities (see Fig 6), which I believe, according to the methodology, are the result of applying the regression models over the wet days. Later on, in Section 4.2, "Performance assessment for regression results", the authors mention that regression models predict the final results presented in this section. By improving the explanation in the methodology, these two sections will be much clearer, and this issue can be solved.

Response: Thank you for your nice suggest. This is indeed a problem that cannot be ignored, we have adjusted the presentation of the results. The results show the performance of the final three merged precipitation (MSMPs: PXGB2, PGBDT2, and PRF2) from different aspects rather than exhibit the performance of classification and regression results separately. We evaluate the MSMPs from the evaluation of precipitation detection ability and precipitation intensity. The title of the Section 4.1 and 4.2 is revised to 4.1 "**Evaluation the precipitation detection ability of MSMPs**" and 4.2 "**Evaluation the precipitation amounts of MSMPs**".   In this way, the advantage of classification and regression models could be well explored and the structure of this section is clearer than before. Meanwhile, we also improving the explanation in the methodology in the revised manuscript.

11- L355: significantly is a statistical loaded term, which must be accompanied by its respective p-value. If the p-value is not provided, I suggest using the word "substantially" instead. Please apply this throughout the manuscript.

Response: Thanks. We have applied "substantially" to replaced "significantly" throughout the manuscript.

12- L395: The authors mention the following "...although Kriging exhibits better performance than original MSPs, its accuracy is strongly dependent on gauge density. This only gauge-based interpolation method would have limited in complex mountainous areas with few gauges." I would be cautious with this statement. Although I agree with it, the ML algorithms cannot predict values outside of their training range, which could be translated into plausible underestimating precipitation over high elevations. Additionally, these techniques are also affected by the size of the training sample. Therefore, the ML techniques, in a sense, have limited performance in complex mountainous areas with few gauges as well.

Response: I very agree with your suggestion. In order to avoid misunderstanding and inappropriate statement. We have reorganized the sentences as follows and revised in manuscript:

although Kriging exhibits better performance than original MSPs, it is only based on gauge observations and does not combine other climate variables associated with precipitation. When MSPs, gauge, and multiple covariates are considered, the MSMPs are more accurate than Kriging.

13- L400: CC or r is also a component of the KGE. Also, see L412. I suggest including it inside of the KGE.

Response: Thank you for your suggestion. We have put CC in the component of the KGE and revised the corresponding statement in manuscript.

14- Figure7: Nice Figure! I think that it should be KGE instead of KEG.

Response: We have revised the KEG to KGE in Figure 7 in the revised manuscript.